



Atmospheric
Measurement
Techniques

# Effects of different correction algorithms on absorption coefficient – a comparison of three optical absorption photometers at a boreal forest site

**Krista Luoma[1], Aki Virkkula[2], Pasi Aalto[1], Katrianne Lehtipalo[1,2], Tuukka Petäjä[1], and Markku Kulmala[1]**

[1]Institute for Atmospheric and Earth System Research, University of Helsinki, Helsinki, 00014, Finland
[2]Atmospheric Composition Research, Finnish Meteorological Institute, Helsinki, 00560, Finland

**Correspondence:** Krista Luoma (krista.q.luoma@helsinki.fi)

**Abstract.** We present a comparison between three absorption photometers that measured the absorption coefficient ($\sigma_{abs}$) of ambient aerosol particles in 2012–2017 at SMEAR II (Station for Measuring Ecosystem–Atmosphere Relations II), a measurement station located in a boreal forest in southern Finland. The comparison included an Aethalometer (AE31), a multi-angle absorption photometer (MAAP), and a particle soot absorption photometer (PSAP). These optical instruments measured particles collected on a filter, which is a source of systematic errors, since in addition to the particles, the filter fibers also interact with light. To overcome this problem, several algorithms have been suggested to correct the AE31 and PSAP measurements. The aim of this study was to research how the different correction algorithms affected the derived optical properties. We applied the different correction algorithms to the AE31 and PSAP data and compared the results against the reference measurements conducted by the MAAP. The comparison between the MAAP and AE31 resulted in a multiple-scattering correction factor ($C_{ref}$) that is used in AE31 correction algorithms to compensate for the light scattering by filter fibers. $C_{ref}$ varies between different environments, and our results are applicable to a boreal environment. We observed a clear seasonal cycle in $C_{ref}$, which was probably due to variations in aerosol optical properties, such as the backscatter fraction and single-scattering albedo, and also due to variations in the relative humidity (RH). The results showed that the filter-based absorption photometers seemed to be rather sensitive to the RH even if the RH was kept below the recommended value of 40 %. The instruments correlated well ($R \approx 0.98$), but the slopes of the regression lines varied between the instruments and correction algorithms: compared to the MAAP, the AE31 underestimated $\sigma_{abs}$ only slightly (the slopes varied between 0.96–1.00) and the PSAP overestimated $\sigma_{abs}$ only a little (the slopes varied between 1.01–1.04 for a recommended filter transmittance > 0.7). The instruments and correction algorithms had a notable influence on the absorption Ångström exponent: the median absorption Ångström exponent varied between 0.93–1.54 for the different algorithms and instruments.

## 1 Introduction

Atmospheric aerosol particles have a notable effect on the Earth's radiative balance. The particles affect the Earth's climate directly by scattering and absorbing radiation from the Sun and indirectly through aerosol–cloud interactions (IPPC, 2013). According to an IPCC report (IPPC, 2013), one of the greatest uncertainties in determining the global radiative forcing is related to atmospheric aerosol particles. Reasons for the large uncertainty are the complex nature of aerosol–cloud interactions and also the great spatiotemporal variation of the particles (Lohmann and Feichter, 2005). Since the number concentration, size distribution, chemical composition, and shape of the particles vary in both space and time, it is challenging to model and estimate the effect that the aerosol particles have on climate on a global scale (IPCC, 2013).

Generally, the direct effect of aerosol particles on climate is cooling since most of the particles scatter radiation from the Sun back into space (IPPC, 2013). However, if particles that are dark (i.e., highly absorbing) are located above a bright surface (i.e., highly scattering), the particles have a warming effect on climate. The sign (i.e., negative sign for the cooling effect and positive sign for the warming effect) of the aerosol forcing efficiency depends on the darkness of the particles, which is described by single-scattering albedo ($\omega$), and on the albedo of the ground below the aerosol layer (Haywood and Shine, 1995). To determine the direct effect of aerosol particles, in addition to the information about the albedo of the surface, we need measurements of aerosol optical properties (AOPs) like scattering, backscattering, and absorption coefficients ($\sigma_{sca}$, $\sigma_{bsca}$, and $\sigma_{abs}$). $\sigma_{sca}$ is a measure of light scattering by the particles in all directions; $\sigma_{bsca}$ is a measure of light scattering only in the backward direction, and $\sigma_{abs}$ is a measure of particulate light absorption. All these variables are wavelength dependent, which is why the measurements of AOPs are preferably conducted at multiple wavelengths.

Measuring $\sigma_{sca}$ and $\sigma_{bsca}$ is rather straightforward, and the measurements are typically conducted with an integrating nephelometer. Correction algorithms and coefficients to minimize the error sources and uncertainties of integrating nephelometers are systemically used (Anderson and Ogren, 1998; Müller et al., 2011b). However, for the $\sigma_{abs}$ measurements there are still large uncertainties and the error sources are not as well defined as for the $\sigma_{sca}$ and $\sigma_{bsca}$ measurements. The main difference between the $\sigma_{sca}$ and $\sigma_{abs}$ measurements is that the $\sigma_{sca}$ measurements are conducted for particles suspended in air, whereas $\sigma_{abs}$ is typically measured by filter-based techniques, where the aerosol particles are collected on a filter. The problem with the filter-based measurements is that in addition to the particles, the filter fibers also interact with radiation and thus influence the measurements.

One of the issues arising specifically with the optical-filter-based measurements is the multiple scattering of light by the filter fibers. The multiple scattering is considered by the so-called multiple-scattering correction factor ($C_{ref}$). Even though $C_{ref}$ should only depend on the properties of the filter, previous studies have shown that $C_{ref}$ also depends on the particulate matter suspended in the filter (Arnott et al., 2005; Collaud Coen et al., 2010; Weingartner et al., 2003). $C_{ref}$ has been observed to vary from station to station, and therefore, it has been studied in different environments. For example, Collaud Coen et al. (2010) studied $C_{ref}$ at very clean mountain sites, in a maritime site, and in urban areas; Schmid et al. (2006) made observations in Amazonia; Backman et al. (2017) studied $C_{ref}$ values in Arctic sites; and Kim et al. (2019) ran measurements in a maritime site, high-altitude sites, and Arctic sites. Since there is no generally accepted method for deriving the $C_{ref}$ values, the methods between different studies vary, which can also affect the results. In this study, we derived $C_{ref}$ by comparing two optical-filter-based instruments with each other.

Another issue with optical-filter-based measurements is related to the nonlinear response of the instruments as the filter is loaded with particles. When the filter is loaded with absorbing particles, the particle loading decreases the response of the instrument. Therefore, the instruments report a lower $\sigma_{abs}$ for loaded filters compared to pristine filter measurements. Several studies have developed algorithms to overcome this problem that has been observed with different instruments (Arnott et al., 2005; Bond et al., 1999; Collaud Coen et al., 2010; Li et al., 2020; Müller et al., 2014; Ogren, 2010; Schmid et al., 2006; Weingartner et al., 2003; Virkkula et al., 2005, 2007; Virkkula, 2010). In general, after correcting the data for the multiple-scattering and loading effects, the absorption instruments agree rather well with the reference measurements (Drinovec et al., 2015; Hyvärinen et al., 2013; Park et al., 2010; Segura et al., 2014). The outcome of the different algorithms, however, varies, and they may affect, for example, the wavelength dependency of $\sigma_{abs}$ (Backman et al., 2014; Collaud Coen et al., 2010).

This study has two aims that address the variation in $C_{ref}$ and the differences between the correction algorithms. The first aim is to provide $C_{ref}$ values that are suitable for a boreal forest site and to study how $C_{ref}$ varies for different correction algorithms. The second aim is to present how the different correction algorithms of $\sigma_{abs}$ affect the measured and derived AOPs.

The measurements presented in this study were conducted in 2012–2017 at the Station for Measuring Ecosystem–Atmosphere Relations II (SMEAR II; Hari and Kulmala, 2005), which is located in the middle of a boreal forest in southern Finland. During this period, the AOPs at SMEAR II were measured by several instruments – an integrating nephelometer and three different absorption photometers (AE31 Aethalometer; particle soot absorption photometer, PSAP; and multi-angle absorption photometer, MAAP) – which enabled determining $C_{ref}$ and an extensive comparison between the different instruments and correction algorithms. AOPs at SMEAR II have been extensively discussed by Virkkula et al. (2011) and Luoma et al. (2019); however, these studies focused on the temporal variation in the AOPs and they only discussed nephelometer and AE31 data. In this study, we focus on the technical side of the measurements and instrument comparison.

## 2 Measurements and methods

### 2.1 The field site

The measurements took place at SMEAR II (Station for Measuring Ecosystem–Atmosphere Relations II; Hari and Kulmala, 2005). The measurement station is located in Hyytiälä, southern Finland (61°51′ N, 24°17′ E). SMEAR II is a rural

measurement station, and it represents a boreal forest environment. The area around the station is mostly forests that consist mainly of Scots pine trees (Hari et al., 2013). The site is classified as rural, and there are no significant sources of pollution nearby. The area is sparsely populated; in the nearby area there are a few smaller towns and some scattered settlements. The closest bigger cities are Tampere (220 000 inhabitants) and Jyväskylä (140 000 inhabitants), and they are located 60 and 100 km away from the station.

## 2.2 Instrument setup

The measurements of AOPs for $PM_{10}$ particles were started in June 2006 with an integrating nephelometer (TSI model 3563) and an Aethalometer (Magee Scientific model AE31). Later on a particle soot absorption photometer (PSAP; Radiance Research 3λ model; Virkkula et al., 2005) and a multi-angle absorption photometer (MAAP; Thermo Scientific model 5012; Petzold and Schönlinner, 2004) were also used.

The measurement arrangement of the instruments that measured the AOPs is presented in Fig. 1. The schematic figure represents the measurement line from a period when all the instruments mentioned before were measuring in parallel, which was during 2014–2015. At the start of the measurement line, a pre-impactor removed all the particles that were larger than 10 μm in aerodynamic diameter (i.e., $PM_{10}$ passed the pre-impactor). The airflow through another impactor, which removed all the particles larger than 1 μm (i.e., $PM_1$ passed the impactor), was controlled by two valves. The valves changed the direction of the flow every 10 min, so in a 20 min measurement cycle the instruments were exposed for 10 min to the $PM_{10}$ and then for 10 min to the $PM_1$. To hinder the effect of changing inlets, the first few minutes of the measurements after the inlet switch were omitted. For the absorption instruments the first 3 min was omitted, and for the integrating nephelometer the first 5 min was omitted. The sample air was dried with Nafion dryers for the PSAP, AE31, and integrating nephelometer for the whole period and for the MAAP from March 2017. Also, a cavity attenuated phase shift (CAPS) extinction monitor (Aerodyne Research; Kebabian et al., 2007) is marked in Fig. 1 since it was part of the measurement line. However, due to technical issues CAPS data were not applied in this study.

Even though the measurements of AOPs have been conducted at SMEAR II since 2006, in this study, we consider only data measured after January 2012 until December 2017. This period was selected to have at least two absorption instruments running in parallel: the AE31 stopped operating in December 2017; the PSAP operated from January 2012 to March 2016, and the MAAP started operation in June 2013. Also, during this period there were only a few changes in the measurement line: in March 2017 the MAAP flow was decreased from 18 to 9 L min$^{-1}$ and Nafion dryers were installed in front of the MAAP, and in November 2017 one

of the two Nafion dryers was removed in front of the nephelometer.

The instruments measured AOPs at different wavelengths: the integrating nephelometer measured at three wavelengths (450, 550, and 700 nm); the AE31, the PSAP, and the MAAP measured at seven wavelengths (370, 470, 520, 590, 660, 880, and 950 nm), three wavelengths (467, 530, and 660 nm), and one wavelength (637 nm), respectively. Here, we report the typically used AE31 and PSAP wavelengths, which are reported in the AE31 manual and by Virkkula et al. (2005), respectively. These reported wavelengths deviate slightly from the ones measured and reported by Müller et al. (2011a) (see their Table 6). For the MAAP, we decided to use the wavelength reported by Müller et al. (2011a) since it more commonly used and it clearly deviated from the wavelength reported by the manual.

The data availability of all the instruments for the studied period sets is reported in Fig. S1 in the Supplement. Some of the data were missing or invalidated due to instrument malfunctions, too-high relative humidity (RH), or the absence of the instrument because of workshops or campaigns. If the RH exceeded 40 % in an instrument, the data were marked as invalid according to recommendations (WMO/GAW, 2016). Before the dryers were installed for the MAAP in March 2017, some of the MAAP data, especially from the summer, were invalidated due to too-high RH. During the cold season, the indoor temperature at the measurement cottage was higher than outdoors, and therefore the RH decreased when the sample air was warmed up to the indoor temperature (passive drying). However, in the summer the RH sometimes increased above the accepted limit since the passive drying was not enough due to minimal difference between the indoor and outdoor temperature.

## 2.3 Absorption measurements

As mentioned above, the $\sigma_{abs}$ of aerosol particles at different wavelengths at SMEAR II was measured with three different instruments: an AE31, PSAP, and MAAP. Each of these instruments measured $\sigma_{abs}$ by a filter-based technique, which means that the measurements were conducted for aerosol particles that were collected on a filter. The AE31 operated on a quartz fiber filter (Pallflex, type Q250F), the PSAP on a quartz fiber filter (Pall, type E70-2075W), and the MAAP on a glass fiber filter (Thermo Scientific, type GF10).

The AE31 and the PSAP have a similar measurement principle (Bond et al., 1999). Before $\sigma_{abs}$ can be determined by using different correction algorithms, the instruments measure the attenuation coefficient ($\sigma_{ATN}$), which is the attenuation of light through the sample collected on the filter. The equation for $\sigma_{ATN}$ is derived from the Beer–Lambert–Bouguer law:

$$\sigma_{ATN} = \frac{A}{Q\,\Delta t} \ln \frac{I_{t-\Delta t}}{I_t} = \frac{A}{Q} \frac{\Delta ATN}{\Delta t}, \tag{1}$$

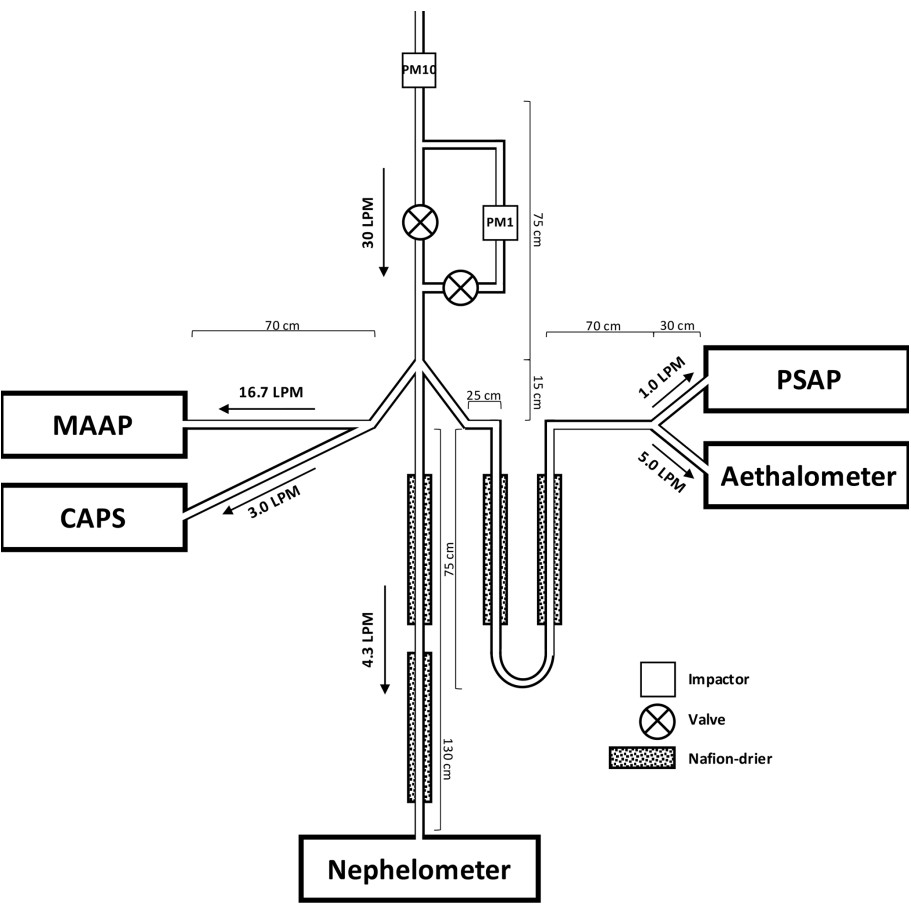

**Figure 1.** Measurement scheme for the instruments that measured the aerosol optical properties at the SMEAR II station. This setup ran during 2014–2015, when all the instruments were operating in parallel.

where $A$ is the sample area on the filter, $Q$ is the flow through the filter, and $\Delta t$ is the length of the measurement period. $I_{t-\Delta t}$ and $I_t$ are the measured and normalized light intensities through the filter at the beginning of the measurement period ($t - \Delta t$) and at the end of the measurement period ($t$). The intensities are normalized by comparing them to the intensity measured through a clean reference spot. Normalizing the intensities accounts for possible drifts and changes in the intensities of the LEDs. $\Delta$ATN is the change in attenuation (ATN), which is calculated from the ratio of light intensity through a clean filter ($I_0$) and through a loaded filter ($I_t$) as

$$\mathrm{ATN} = -\ln\left(\frac{I_t}{I_0}\right) \cdot 100\,\%. \qquad (2)$$

In addition to ATN, the filter loading can also be described by transmittance (Tr)

$$\mathrm{Tr} = I_t I_0^{-1}, \qquad (3)$$

which can be also presented as a function of ATN (Tr = exp(ATN/100 %)). ATN and Tr represent essentially the same concept, but the way of expressing the change in in-

tensity depends on the instrument used: ATN is traditionally associated with Aethalometer data and Tr with PSAP data.

In Eq. (1), $A$ is typically a constant value defined by the manufacturer and $Q$ is recorded and reported by the instrument. These values, however, might deviate notably from the real values, and therefore they should be measured and checked regularly. If these values differ from the reported ones, Eq. (1) needs corrections for $A$ and $Q$. At SMEAR II, the sample flow of each instrument was regularly measured with a Gilian flow meter, and the $Q$ reported by the instruments was corrected to match the Gilian measurements. For the PSAP and AE31 we used the $A$ values of 18.1 and 54.8 mm$^2$, which deviated from the default ones of 17.8 and 50.0 mm$^2$, respectively. The $A$ used by default in the MAAP matched the measured one, and therefore it was not corrected.

In a filter, the light is attenuated because of the absorption and scattering by the particles but also because of the scattering by the filter fibers, which is called multiple scattering. The scattering by the filter fibers increases the optical path of the light beam through the filter. Therefore, the probability of the light beam being absorbed by a particle

increases. Because of the scattering in the filter medium, $\sigma_{\text{ATN}}$ is larger than $\sigma_{\text{abs}}$. Not only do the filter fibers scatter light, but also the embedded aerosol particles scatter light and cause so-called apparent absorption, which is typically considered by subtracting a fraction of scattering from $\sigma_{\text{ATN}}$. In addition to the scattering by the fibers and particles, the increasing number of absorbing particles in the filter also affects the instrumental response. The signal response caused by the particulate absorption decreases with increasing filter loading. Absorbing particles induce a so-called shadowing or a loading effect, which decreases the change in the intensity ($I_{t-\Delta t} I_t^{-1}$) as the filter becomes more loaded (Weingartner et al., 2003). This means that the instrumental response is nonlinear with increasing filter loading. The increasing filter loading has an opposite effect to the scattering of the filter fibers and particles: the absorbing particles collected on the filter decrease the optical path, and therefore the reported $\sigma_{\text{ATN}}$ for a loaded filter is lower than for a pristine filter. This nonlinearity is considered in the various correction algorithms presented in Sect. 2.3.1 and 2.3.2.

The measurement principle of the MAAP is different from that of the AE31 and PSAP (Petzold and Schönlinner, 2004). In addition to the light attenuation measurements, the MAAP also measures the backscattered light from the filter at two different angles. $\sigma_{\text{abs}}$ is then obtained by using a radiative transfer scheme where the measurements of the backscattering and light attenuation are taken into account (Petzold and Schönlinner, 2004). Because of the backscattering measurements, the MAAP does not suffer as much from the filter artifacts as the Aethalometer and the PSAP. However, in very polluted environments the MAAP also suffers from a measurement artifact that has to be corrected (Hyvärinen et al., 2013), which at SMEAR II, however, is not the case.

At SMEAR II, the MAAP advanced the filter spot automatically once per day in 24 h intervals. The AE31 also advanced the spot automatically when ATN reached 120 at a 370 nm wavelength. The PSAP filters were changed manually, and the aim was to change the filter every second day, but due to weekends and holidays, the filters were sometimes changed only after several days. On average the PSAP filters were changed once every 3 d.

The reported uncertainties in the MAAP, PSAP, and Aethalometer are 12 %, 13 %, and as large as 50 %, respectively (Arnott et al., 2005; Ogren, 2010; Petzold and Schönlinner, 2004). Müller et al. (2011a) reported that the unit-to-unit variability in the PSAP, AE31, and MAAP was about 8 %, 20 %, and 3 %. It must be noted that the unit-to-unit variability is a lot smaller, about 2 %, for the new AE33 model (Cuesta-Mosquera et al., 2021). Since the uncertainty and unit-to-unit variability in the MAAP were a lot smaller than for the PSAP and AE31, we used the MAAP as the reference instrument for measuring $\sigma_{\text{abs}}$. However, even though the MAAP was used as the reference here, it must be remembered that like all the filter-based photometers, the MAAP also suffers from the cross sensitivity to purely scattering

aerosols, and therefore it is not the best reference instrument (Müller et al., 2011a).

Each of the absorption photometers used in this study has its strengths and weaknesses that determine which instrument is the most useful in different situations. According to the uncertainty and unit-to-unit variability, the MAAP is the most precise instrument for monitoring $\sigma_{\text{abs}}$ and black carbon (BC) concentration, which is typically derived from $\sigma_{\text{abs}}$ measurements. Also, the backscattering measurements from the filter reduce the artifacts caused by the scattering aerosol particles and the filter-loading effect, making it a more accurate instrument. The MAAP changes the spot in a filter roll automatically, and therefore it does not require much assistance from the operator, and the instrument can run at a remote station as well. However, it measures $\sigma_{\text{abs}}$ only at one wavelength, so it is not possible to perform the source apportionment or interpretation of the chemical composition of the absorbing particles, which requires measurements on several wavelengths (see Sect. 3.1 and Eq. 16). The AE31 has a very wide range of wavelengths, which makes the seven-wavelength Aethalometers, the AE31 and the new model AE33, widely used instruments. Like the MAAP, the AE31 also operates the filter roll automatically, so the instrument does not need that much assistance from the operator. Unfortunately, the problems with defining the errors caused by the filter material are not that well defined, and the instrument uncertainty and unit-to-unit variability in the AE31 are large. The uncertainty in and noise of the PSAP is smaller than those of the AE31, which makes the PSAP a popular instrument especially in areas with low concentrations. Even though the wavelength range is not as wide as with the AE31, the PSAP measures $\sigma_{\text{abs}}$ at three wavelengths, allowing the use of applications that need the wavelength dependency of $\sigma_{\text{abs}}$. The PSAP filters have to be changed manually by the user, so the instrument is not the best option to deploy at a remote site, but then again the leakage through the filter tape is less than for the MAAP and AE31.

### 2.3.1 AE31 correction algorithms

To determine $\sigma_{\text{abs}}$ from AE31 measurements, $\sigma_{\text{ATN}}$ needs to be corrected for the multiple scattering by the filter fibers and for the error caused by the filter loading, and in addition, the scattering of aerosol particles should also be taken into account:

$$\sigma_{\text{abs}} = \frac{\sigma_{\text{ATN}} - a_{\text{s}} \sigma_{\text{sca}}}{C_{\text{ref}} R(\text{ATN})}. \tag{4}$$

The effect of the multiple scattering is corrected with a multiple-scattering correction factor ($C_{\text{ref}}$), and it is larger than unity. For the filter-loading correction ($R(\text{ATN})$) there are different kinds of correction algorithm developed for example by Weingartner et al. (2003), Arnott et al. (2005), Schmid et al. (2006), Virkkula et al. (2007), and Collaud Coen et al. (2010). $R$, which equals unity for unloaded filters, is less than unity for loaded filters, depending on the

filter loading, i.e., ATN defined in Eq. (2). $R$ can also depend on other factors, such as $\omega$, and some of the algorithms also take parameters other than ATN into account.

In Eq. (4), the scattering by the aerosol particles is considered by subtracting a fraction ($a_s$) of the measured scattering ($\sigma_{sca}$). However, the algorithms by Weingartner et al. (2003) and Virkkula et al. (2007) ignore the particle-scattering subtraction, which makes it possible to apply the correction algorithms without any $\sigma_{sca}$ measurements. In the algorithm of Weingartner et al. (2003), however, $\sigma_{sca}$ is considered without the subtraction, as will be shown below. For a comparison, in this study we also present data that were corrected only for the multiple scattering and not for the filter loading (i.e., $R = 1$) or scattering by the particles. Below we present the different algorithms determined by Weingartner et al. (2003), Arnott et al. (2005), Virkkula et al. (2007), and Collaud Coen et al. (2010), which were selected for use in this study.

The current recommendation by the WMO (World Meteorological Organization) and GAW (Global Atmosphere Watch) is to assume $R(\text{ATN})$ is unity for the AE31 and to use a $C_{ref}$ value of 3.5, which was determined by a comparison study of different AE31 instruments (WMO/GAW, 2016). Therefore, we also studied "non-corrected" AE31 data for which we did not apply any $R(\text{ATN})$ correction or particulate scattering reduction but only the multiple-scattering correction.

Weingartner et al. (2003) derived an empirical correction algorithm (hereafter referred to as W2003 and with a subscript WEI) based on laboratory measurements of mixed particles (soot, diesel exhaust, organic coating, ammonium sulfate). The W2003 correction algorithm interpolates the measurements at higher ATN values, to a point where ATN is 10 %. When ATN is lower than 10 %, $R$ is assumed to be unity. In W2003, the loading correction ($R_{WEI}$) is

$$R_{WEI}(\text{ATN}) = \left(\frac{1}{f} - 1\right)\frac{\ln(\text{ATN}) - \ln(10\,\%)}{\ln(50\,\%) - \ln(10\,\%)} + 1. \qquad (5)$$

Weingartner et al. (2003) stated that $R$ depends on the single-scattering albedo ($\omega$), and they found the following relation for the factor $f$:

$$f = a(1 - \omega) + 1. \qquad (6)$$

In Eq. (6), $f$ is unity (i.e., $R$ is unity) when $\omega$ is unity (i.e., the aerosol is purely scattering). Weingartner et al. (2003) determined that $a$ in Eq. (6) was 0.87 and 0.85 at 450 and 660 nm, respectively. According to these two values, we interpolated $a$ for all seven wavelengths by assuming a linear wavelength dependency. Also, $\omega$ was interpolated to the seven AE31 wavelengths according to the mean $\sigma_{abs}$, $\sigma_{sca}$, and scattering Ångström exponent ($\alpha_{sca}$) values reported by Luoma et al. (2019; see their Table 1) for PM$_{10}$ particles. Using these values, we estimated $f$ separately for each wavelength, and we used those constant values in the correction

values. The resulting $a$, $\omega$, and $f$ were slightly wavelength dependent, and their values are presented in Table 1.

The correction algorithm does not apply the scattering correction by subtraction, so $a_{s,WEI} = 0$, and therefore parallel scattering measurements are in principle not needed. However, the effect of the particulate scattering is considered in $f$ since it depends on $\omega$. If there are no parallel measurements of $\sigma_{sca}$, $\omega$ cannot be determined. If there is no estimation of $\omega$, typically $f$ values for different aerosol types determined by Weingartner et al. (2003) are used. The $f$ values were close to the result Weingartner et al. (2003) acquired from measurements of ambient aerosols in a high alpine site and in a garage ($f$ was 1.03 and 1.14 for a "white light" Aethalometer, AE10). For example, Collaud Coen et al. (2010) estimated an intermediate value of $f = 1.10$ for the Cabauw measurement site based on the study by Weingartner et al. (2003).

Arnott et al. (2005) suggested a correction algorithm, which is hereafter referred to as A2005 and with the subscript ARN, based on a well-defined theoretical basis. One big difference from W2003 is that there is a factor for scattering subtraction. Arnott et al. (2005) determined the scattering subtraction fraction $a_{s,ARN}$ from laboratory measurements using submicron ammonium sulfate particles, and the values for different wavelengths are presented in Table 1; however, Arnott et al. (2005) noted that the values of $a_{s,ARN}$ could be different if supermicron aerosol particles were present. The loading correction $R_{ARN}$ was defined as

$$R_{ARN} = \left(\sqrt{1 + \frac{\left(\frac{V\Delta t}{A}\right)\sum_{i=1}^{n-1}\sigma_{abs,i}}{\tau_{a,fx}(\lambda)}}\right)^{-1}, \qquad (7)$$

where $n$ indicates the $n$th measurement after a filter spot change. The correction takes into account the cumulative $\sigma_{abs}$ of the particles collected on the filter material. $\tau_{a,fx}(\lambda)$ is the filter absorption optical depth for the filter fraction $x$ that has particles embedded. To calculate $\tau_{a,fx}(\lambda)$, we used the same power law function $\tau_{a,fx}(\lambda) = \tau_{a,fx,521} \cdot (\lambda/521\,\text{nm})^{-0.754} = 0.2338 \cdot (\lambda/521\,\text{nm})^{-0.754}$. $\tau_{a,fx}$ as Virkkula et al. (2011), and the resulting values are presented in Table 1. The exponent $-0.754$ was obtained from a power function fitting to $\tau_{a,fx}$ vs. $\lambda$ (Table 1 of Arnott et al., 2005), similarly to in Virkkula et al. (2011). $\tau_{a,fx,521} = 0.2338$ is the recommended $\tau_{a,fx}$ value for ambient measurements at 521 nm (Arnott et al., 2005).

Virkkula et al. (2007) proposed a correction algorithm, which is hereafter referred to as V2007 and with the subscript VIR, that utilizes the so-called compensation parameter ($k$). $k$ is determined by comparing the last measurements of a loaded filter to the first measurements conducted with a pristine filter. The compensation parameter is determined for each filter spot (fs) as follows:

$$k_{fs} = \frac{\sigma_{ATN}(t_{fs+1,first}) - \sigma_{ATN}(t_{fs,last})}{\left(\begin{array}{c}\text{ATN}(t_{fs,last})\sigma_{ATN}(t_{fs,last}) \\ -\text{ATN}(t_{fs+1,first})\sigma_{ATN}(t_{fs+1,first})\end{array}\right)}, \qquad (8)$$

**Table 1.** All the wavelength-dependent coefficients used in the AE31 correction algorithms proposed by Weingartner et al. (2003) and Arnott et al. (2005). The extrapolated values of the multiple-scattering correction factor used in the Arnott et al. (2005) correction algorithm ($C_{\mathrm{ARN}}$) at different wavelengths.

| Coefficients for AE31 correction algorithm by Weingartner et al. (2003) | | | | | | |
|---|---|---|---|---|---|---|
| $\lambda$ (nm) | 370 | 470 | 520 | 590 | 660 | 880 | 950 |
| $a$ | 0.878 | 0.868 | 0.863 | 0.857 | 0.850 | 0.829 | 0.822 |
| $\omega$ | 0.91 | 0.89 | 0.89 | 0.88 | 0.85 | 0.83 | 0.82 |
| $f$ | 1.079 | 1.096 | 1.095 | 1.103 | 1.128 | 1.141 | 1.148 |

| Coefficients for AE31 correction algorithm by Arnott et al. (2005) | | | | | | |
|---|---|---|---|---|---|---|
| $\lambda$ (nm) | 370 | 470 | 520 | 590 | 660 | 880 | 950 |
| $100 \cdot a_{\mathrm{s,ARN}}$ | 3.35 | 4.57 | 5.23 | 6.16 | 7.13 | 10.38 | 11.48 |
| $\tau_{\mathrm{a,fx}}$ | 0.3026 | 0.2527 | 0.2338 | 0.2129 | 0.1956 | 0.1575 | 0.1486 |
| $C_{\mathrm{ARN}}$ | 2.70 | 2.82 | 2.87 | 2.94 | 3.00 | 3.16 | 3.20 |

where "first" refers to the mean of the first three values in a pristine filter (i.e., fs + 1) and "last" refers to the mean of the last three values in a loaded filter (i.e., fs). $k$ is then applied to the loading correction $R_{\mathrm{VIR}}$:

$$R_{\mathrm{VIR}}(\mathrm{ATN}) = (1 + k_{\mathrm{fs}}\mathrm{ATN})^{-1}. \tag{9}$$

This algorithm does not take into account the scattering correction, so $a_{\mathrm{s,VIR}} = 0$.

Collaud Coen et al. (2010) applied this correction to data from several stations in Europe and found that it was highly nonstable and that it leads to large outliers. They correctly stated that the difficulty of applying this correction is due to the naturally high variability in $\sigma_{\mathrm{ATN}}$ as a function of time, which is for most of the time greater than the $\sigma_{\mathrm{ATN}}$ decrease induced by filter changes. We therefore calculated the running-average compensation parameter for all seven wavelengths in order to minimize these problems. Then we applied this averaged compensation parameter to correct the non-corrected AE31 data. In other words, the AE31 data were not averaged at this stage, just the compensation parameter.

We determined $k$ as a 14 d running mean ($\pm 7$ d around the changing time of the filter spot), since without the averaging, $k$ was very noisy (see time series for the non-averaged and averaged $k$ in Fig. S6 in the Supplement). Averaging $k$ was also recommended by Virkkula et al. (2007). On average, the 14 d periods included about nine data points (i.e., the filter spot was changed once every 1.6 d). Virkkula et al. (2015) used a similar approach for AE31 data from Nanjing, China, and calculated 24 h running averages of $k$ including on average six filter spot changes.

The Collaud Coen et al. (2010) correction algorithm, which is hereafter referred to as CC2010 and with the subscript COL, was based on the W2003 algorithm, but here the reference ATN for the clean filter is 0 % instead of 10 %. Collaud Coen et al. (2010) determined the $a$ used in Eq. (6) a little differently and obtained a mean value of $a = 0.74$ over different wavelengths and different experiments. $R_{\mathrm{COL}}$ is defined as

$$R_{\mathrm{COL}}(\mathrm{ATN}) = \left(\frac{1}{a\left(1 - \overline{\omega}_{0,n}\right) + 1} - 1\right) \cdot \frac{\mathrm{ATN}}{50\,\%} + 1. \tag{10}$$

Here $\overline{\omega}_{0,\mathrm{fs},n}$ stands for the mean $\overline{\omega}_0$, which was calculated for the filter spot from the first measurements to the $n$th measurement. $\overline{\omega}_{0,\mathrm{fs},n}$ was determined by using $\sigma_{\mathrm{ATN}}$ as the first estimate of $\sigma_{\mathrm{abs}}$. CC2010 also differs from W2003 by considering the scattering correction. Collaud Coen et al. (2010) suggested two kinds of way to determine $a_{\mathrm{s,COL}}$, and here we present the one that was determined in a similar manner to that in A2005. The difference from $a_{\mathrm{s,ARN}}$ is that $a_{\mathrm{s,COL}}$ is determined from the ambient scattering measurements, so it is not constant. $a_{\mathrm{s,COL}}$ is defined similarly to in Arnott et al. (2005) (Eq. 8 in their article), but here the authors used measured scattering properties instead of constant values determined by laboratory measurements:

$$a_{\mathrm{s,COL}} = \overline{\beta}_{\mathrm{sca},n}^{d-1} c\lambda^{-\overline{\alpha}_{\mathrm{sca},n}(d-1)}, \tag{11}$$

where $d = 0.564$ and $c = 0.329 \times 10^{-3}$. In Eq. (11) the overlined variables, $\alpha_{\mathrm{sca},n}$ and $\beta_{\mathrm{sca},n}$, stand for average properties of aerosols deposited in the filter, i.e., mean values from the beginning of the filter measurements until the $n$th measurement. $\alpha_{\mathrm{sca},n}$ is the scattering Ångström exponent (see Sect. 3.1 and Eq. 16), and $\beta_{\mathrm{sca}}$ is acquired from the power law fit of the wavelength dependency of $\sigma_{\mathrm{sca}}$:

$$\sigma_{\mathrm{sca}} = \beta_{\mathrm{sca}}\lambda^{-\alpha_{\mathrm{sca}}}, \tag{12}$$

where the fit is calculated with $\lambda$ and $\sigma_{\mathrm{sca}}$ in units of nanometers (nm) and inverse megameters (Mm$^{-1}$) to acquire unitless $\beta$.

### 2.3.2 PSAP correction algorithms

Since the measurement principles of the PSAP and AE31 are basically the same, the PSAP data need similar kinds of correction to the AE31 data (Eq. 2). In this study the PSAP data

were corrected with two algorithms: one described by Bond et al. (1999) and later specified by Ogren (2010), which is hereafter referred to as B1999, and the other determined by Virkkula et al. (2005) and later corrected by Virkkula (2010), which is hereafter referred to as V2010. The algorithms of Müller et al. (2014) and Li et al. (2020) were not applied.

The B1999 correction algorithm revisited by Ogren et al. (2010) is given by

$$\sigma_{abs} = f(Tr)\sigma_0 - a_s\sigma_{sca} , \qquad (13)$$

where

$$f_{Tr} = (1.5557 \cdot Tr + 1.0227)^{-1} . \qquad (14)$$

In the V2010 correction algorithm, $\sigma_{PSAP}$ is determined in an iterative manner. The first estimation of the absorption coefficient ($\sigma_{abs,0}$) is determined by $\sigma_{abs,0} = (k_0 + k_1 \ln(Tr))\sigma_{ATN} - s\sigma_{sca}$, where $k_0$ and $k_1$ are constants presented in Table 1 in Virkkula (2010). $\sigma_{abs,0}$ is used to calculate the single-scattering albedo $\omega$ (see Sect. 3.1 and Eq. 17), which is then again used to calculate $\sigma_{abs}$, again in an iterative manner but now with a different kind of equation:

$$\sigma_{abs} = (k_0 + k_1 h(\omega_0) \ln(Tr))\sigma_{ATN} - s\sigma_{sca} , \qquad (15)$$

where $h(\omega_0) = h_0 + h_1\omega$. $\omega$ is then calculated again with Eq. (17). These two steps are repeated until the change in $\sigma_{abs}$ is minor. Here, the iteration was stopped once the change was less than 1 %. It must be noted that this correction algorithm is different from V2007 determined for the Aethalometer data.

### 2.3.3 Differences between the algorithms

The W2003 algorithm only depends on ATN; otherwise it applies constant values, and it does not consider the scattering subtraction. A2005 is not a function of ATN, but it takes the filter loading into account by summing the $\sigma_{abs}$ values of the accumulated particles on the filter spot. It does not assume a constant for the scattering reduction but determines the fraction from the wavelength dependency of $\sigma_{sca}$. The CC2010 algorithm is similar to that of A2005 in the sense that it also defines its own scattering reduction factor and determines the filter-loading correction by taking into account the properties of the particles accumulated in the filter. V2007 only depends on the difference between the last and first measurements of two filter spots, and it assumes no constant coefficients. The B1999 algorithm relies heavily on constants that describe the dependency on Tr, whereas the V2010 algorithm is an iterative process that depends on $\omega$. Both B1999 and V2010 consider the scattering reduction with a coefficient.

### 2.4 Scattering measurements

The $\sigma_{sca}$ data are needed to subtract a fraction of particulate scattering from the $\sigma_{ATN}$ values in A2005, CC2010, B1999,

and V2010. Measurements of $\sigma_{sca}$ and $\sigma_{bsca}$ are also needed in determining $\omega$ and the backscatter fraction ($b$; see Sect. 3.1 and Eq. 18), which are used to explain the observed variations in the results. $\omega$ is also used in CC2010.

$\sigma_{sca}$ and $\sigma_{bsca}$ were measured with an integrating nephelometer (TSI model 3565; Anderson et al., 1996). The integrating nephelometer measured $\sigma_{sca}$ and $\sigma_{bsca}$ at three wavelengths (450, 550, and 700 nm). Due to instrumental restrictions, the nephelometer can only measure $\sigma_{sca}$ in the range of 7–170° and $\sigma_{bsca}$ in the range of 90–170°, and therefore a truncation correction is applied to $\sigma_{sca}$ and $\sigma_{bsca}$ measurements (Anderson and Ogren, 1998; Bond et al., 2009). The fractional uncertainty in the integrating nephelometer for $PM_{10}$ has been reported to be $\pm9$ % (Sherman et al., 2015). Since scattering by aerosol particles depends significantly on the particles' size, the particulate light scattering is sensitive to hygroscopic growth. To prevent this, the integrating nephelometer is operated with two Nafion dryers as shown in Fig. 1.

## 3 Data analysis

All the data were averaged to 1 h intervals. The $PM_1$ and $PM_{10}$ measurements were not separated in the data analysis, and the $PM_1$ and $PM_{10}$ data were averaged together. Since all the instrument measured the same sample air, combining the $PM_1$ and $PM_{10}$ data caused no discrepancies between the instruments. Since this study discusses filter-based absorption photometers and ATN in the filters decreases due to the accumulation of both $PM_1$ and $PM_{10}$, it would have been difficult to separate the effect of the different size cuts in the data analysis, and therefore the data of different size cuts were combined.

### 3.1 Intensive properties

The intensive properties of aerosol particles are determined from the measurements of the extensive properties, which in our data are $\sigma_{abs}$, $\sigma_{sca}$, and $\sigma_{bsca}$. In addition to the chemical properties and size distribution, the extensive properties also depend on the number and volume concentration of particles. The intensive properties, however, are independent of the number of particles, and they depend only on the properties of the particles, such as the shape of the size distribution, chemical composition, and shape of the particles. Therefore, intensive properties are useful parameters as they indirectly indicate the properties of the particle population. The intensive properties used in this article are the Ångström exponent ($\alpha$), single-scattering albedo ($\omega$), and backscatter fraction ($b$), and they are presented below.

The Ångström exponent ($\alpha$) describes the wavelength dependency of the optical properties, and it can be calculated for example for $\sigma_{abs}$ and $\sigma_{sca}$ to acquire the absorption Ångström exponent ($\alpha_{abs}$) and scattering Ångström exponent

$(\alpha_{\mathrm{sca}})$, respectively. $\alpha$ is defined by

$$\alpha = -\frac{\ln \frac{\sigma_1}{\sigma_2}}{\ln \frac{\lambda_1}{\lambda_2}}, \tag{16}$$

where $\sigma_1$ and $\sigma_2$ are the property for which $\alpha$ is calculated at wavelengths $\lambda_1$ and $\lambda_2$, respectively. $\alpha$ is typically used to interpolate or extrapolate optical properties to different wavelengths. This is useful for example in cases when instruments measure optical properties at different wavelengths and the measurements between different instruments need to be compared. The wavelength dependency also gives information about the size distribution, chemical composition, and sources of the particles: $\alpha_{\mathrm{sca}}$ depends on the size distribution of the particles, and $\alpha_{\mathrm{abs}}$ depends on both the chemical composition and the size distribution. $\alpha_{\mathrm{abs}}$ is typically used in a set of empirical equations that approximate the source of black carbon (BC) (Sandradewi et al., 2008; Zotter et al., 2017).

The single-scattering albedo ($\omega$) describes how big the fraction of the total light extinction ($\sigma_{\mathrm{ext}} = \sigma_{\mathrm{abs}} + \sigma_{\mathrm{sca}}$) is due to scattering:

$$\omega = \frac{\sigma_{\mathrm{sca}}}{\sigma_{\mathrm{sca}} + \sigma_{\mathrm{abs}}}. \tag{17}$$

The lower $\omega$ is, the darker the aerosol particles are, which is typically caused by a higher content of black carbon (BC); $\omega$ close to unity indicates that the particles are high in scattering material like sulfates or sea salt. Therefore, $\omega$ is a rough indicator of the chemical composition of the particles.

The backscatter fraction ($b$) describes the fraction of backscattering coefficient ($\sigma_{\mathrm{bsca}}$; meaning that the light scatters in the backward hemisphere) of the total scattering coefficient ($\sigma_{\mathrm{sca}}$):

$$b = \frac{\sigma_{\mathrm{bsca}}}{\sigma_{\mathrm{sca}}}. \tag{18}$$

$b$ is also size dependent. In the molecular size range it is 0.5, which means that the particles scatter light evenly in the forward and in the backward direction. For larger particles $b$ decreases, so the particles scatter light more in the forward direction.

### 3.2 Multiple-scattering correction factor

As stated by Weingartner et al. (2003), $C_{\mathrm{ref}}$ should in principle only depend on the instrument and the filter material used. The effect caused by different numbers of particles deposited in the filter material and their optical properties should be taken into account by the empirical filter-loading correction functions $R(\mathrm{ATN})$. However, as shown by previous studies, $C_{\mathrm{ref}}$ varies both spatially and temporally (Backman et al., 2017; Collaud Coen et al., 2010), and therefore we also determined $C_{\mathrm{ref}}$ at SMEAR II.

In this study, the multiple-scattering correction factor ($C_{\mathrm{ref}}$) was defined for the AE31 measurements by using the

$\sigma_{\mathrm{abs}}$ measured by the MAAP as the reference absorption coefficient ($\sigma_{\mathrm{abs,ref}}$). To determine $C_{\mathrm{ref}}$, the $\sigma_{\mathrm{ATN}}$ measured by the AE31 had to be corrected for the artifact caused by the increased filter loading, and then the measurements could be compared to the reference absorption ($\sigma_{\mathrm{abs,ref}}$) measured by the MAAP:

$$C_{\mathrm{ref}} = \frac{\sigma_{\mathrm{ATN}}}{R(\mathrm{ATN})\sigma_{\mathrm{abs,ref}}}. \tag{19}$$

$C_{\mathrm{ref}}$ was defined separately for data corrected using W2003, A2005, V2007, and CC2010 to obtain $C_{\mathrm{WEI}}$, $C_{\mathrm{ARN}}$, $C_{\mathrm{VIR}}$, and $C_{\mathrm{COL}}$, respectively. $C_{\mathrm{ref}}$ was also determined for data that were not corrected for the filter loading ($C_{\mathrm{NC}}$, where subscript NC stands for non-corrected). Because the MAAP measures $\sigma_{\mathrm{abs,ref}}$ only on the 637 nm wavelength, the closest AE31 and nephelometer data were first interpolated to the same wavelength. $\sigma_{\mathrm{ATN}}$, ATN, and $\sigma_{\mathrm{sca}}$ were interpolated to 637 nm by applying the Ångström exponent explained in Eq. (16). Also, the wavelength-dependent constants used in W2003 and A2005 were interpolated to 637 nm. The $f$ used in W2003 at 637 nm was 1.12, and the $a_{\mathrm{s,ARN}}$ and $\tau_{\mathrm{a,fx}}$ used in A2005 were 0.0681 and 0.2009, respectively, at 637 nm.

In A2005, cumulative optical properties of the particles collected on the filter were needed, and thus $C_{\mathrm{ARN}}$ was determined by iteration; $C_{\mathrm{ARN}}$ was iterated for each filter spot until the median $\sigma_{\mathrm{abs,ref}}$ and $\sigma_{\mathrm{abs,ARN}}$ agreed within a 1 % limit. Because of the iteration, there is one $C_{\mathrm{ARN}}$ value for each filter spot. For other correction algorithms, the $C_{\mathrm{ref}}$ value was determined by two methods: (1) as the slope of a linear regression for the whole data set (linear fit for a loading-corrected $\sigma_{\mathrm{ATN}}$-vs.-$\sigma_{\mathrm{ref}}$ plot; Eq. 19) and (2) by simply using Eq. (19) to acquire the $C_{\mathrm{ref}}$ value for each measurement point separately. In A2005, $C_{\mathrm{ARN}}$ depends on the wavelength. In this study $C_{\mathrm{ARN}}$ was determined only at 637 nm. Since we followed a similar procedure to that presented by Arnott et al. (2005), a fraction of $\sigma_{\mathrm{sca}}$ was first subtracted from $\sigma_{\mathrm{ATN}}$ before determining $C_{\mathrm{ARN}}$, which is different to Eq. (19).

## 4 Results and discussion

### 4.1 Multiple-scattering correction for the AE31

The different $C_{\mathrm{ref}}$ values were determined by a linear fit by comparing loading-corrected AE31 data to the reference data from the MAAP. Since $C_{\mathrm{ref}}$ is described only by the slope of the fit, the intercept on the $y$ axis of the fit was forced to be zero. For the linear fit we used all the available parallel data from the AE31 and MAAP. The $C_{\mathrm{ref}}$ values were 3.00, 3.14, 2.99, and 2.77 for data corrected by W2003, V2007, and CC2010 and for data that were not corrected, respectively. The results and their statistical variability are presented in Table 2. The relatively small standard error (SE) and the range of confidence intervals (CI) indicate that the differences between the $C_{\mathrm{ref}}$ values were statistically significant. However,

for example, the difference between $C_{WEI}$ and $C_{COL}$ was small.

The smallest determined $C_{ref}$ value was $C_{NC}$, which was expected. Since $\sigma_{ATN}$ decreases for a loaded filter and the filter-loading correction was not applied, $C_{NC}$ had to be smaller than for the corrected data. Since the values of $C_{WEI}$ and $C_{COL}$ were almost the same, the result suggested that the loading corrections $R_{WEI}$ and $R_{COL}$ had on average a similar effect on the data. The highest value was determined for $C_{VIR}$, which suggests that on average, the value of $R_{VIR}$ was the lowest (i.e., the effect of filter-loading correction in V2007 was stronger).

Since $C_{ARN}$ was determined in an iterative manner for each filter spot, $C_{ARN}$ was calculated as the median of all the filter spots and the resulting value was 3.13, which is also shown in Table 2. This result is not directly comparable to the other $C_{ref}$ values that were derived as a linear fit. Also, unlike the other algorithms, A2005 assumed a wavelength-dependent $C_{ARN}$. Here, we were only able to determine $C_{ref}$ at one wavelength by comparing the interpolated AE31 data to the MAAP measurements at 637 nm. To acquire $C_{ARN}$ at all seven wavelengths of the AE31, we used the power law function $C_{ARN}(\lambda) = C_{ARN,637\,nm}(\lambda/637\,nm)^{0.181} = 3.13 \cdot (\lambda/637\,nm)^{0.181}$, where the exponent 0.181 was obtained from a power function fitting to $C_{ref}$ vs. $\lambda$ in Table 1 of Arnott et al. (2005), similarly to Virkkula et al. (2011). $C_{ARN,637\,nm} = 3.13$ is the value determined above at $\lambda = 637\,nm$. The results of the wavelength-dependent $C_{ARN}$ values are presented in Table 3.

According to Collaud Coen et al. (2010), who studied the $C_{ref}$ values of different algorithms for ambient measurements in different environments, the higher $C_{ref}$ values were typically measured in polluted areas. Observations in our study support this claim. For example, the authors determined a mean $C_{WEI}$ of 2.81, 2.81, 3.05, and 4.09 at Hohenpeißenberg, Jungfraujoch, Mace Head, and Cabauw, respectively. Segura et al. (2014) obtained a $C_{ref}$ value of 4.22 measured in Granada, Spain, at 637 nm for the correction algorithm by Schmid et al. (2006). Compared to their study the $C_{ref}$ values at SMEAR II were obviously lower than the mean $C_{ref}$ values at the Cabauw and Granada measurement stations. Cabauw station is located near populated and industrial areas, and the station in Granada is located close to a highway. At SMEAR II, the average $C_{ref}$ values were somewhat higher than in the clean mountain stations in Hohenpeißenberg and Jungfraujoch. The closest values were defined for the Mace Head station, which observes mostly marine air. The $C_{ref}$ values by Collaud Coen et al. (2010) and Segura et al. (2014) were determined similarly to in our study, by comparing AE31 measurements against those of the MAAP.

Backman et al. (2017) determined $C_f$ (Backman et al., 2017, used the symbol $C_f$ instead of $C_{ref}$ to mark that the comparison was not conducted with a reference instrument) values for ambient data at several Arctic sites. They also derived $C_f$ optically by comparing Aethalometer measurements against those of a MAAP, PSAP, and CLAP (continuous light absorption photometer; Ogren et al., 2017) They ran the comparison for Aethalometer data that were not corrected for the filter-loading error. The median $C_f$ values at 637 nm were 1.61, 3.12, 3.42, 4.01, and 4.22 measured at Summit, Barrow, Alert, Tiksi, and Pallas, respectively. Backman et al. (2017) did not find any clear explanation for the very low $C_f$ at Summit. At the other sites, the $C_f$ values were rather high compared to the $C_{NC}$ observed at SMEAR II ($C_{NC} = 2.77$), which is unexpected if we assume that $C_{ref}$ was lower in clean environments, such as the Arctic, compared to at sites closer to pollution sources.

In laboratory runs, Arnott et al. (2005) determined $C_{ref} = 2.076$ (at 521 nm) for kerosene soot by comparing an AE31 against a photoacoustic instrument, and Weingartner et al. (2003) observed $C_{ref} = 2.14$ (averaged over wavelengths) for non-coated soot particles by subtracting scattering from extinction measurements. Compared to the $C_{ref}$ determined in laboratory studies by Weingartner et al. (2003) and Arnott et al. (2005), the ambient measurements in our study yielded higher values. This was also observed by Arnott et al. (2005), who suggested $C_{ref} = 3.688$ (at 521 nm) for ambient measurements, which is closer to our observations. In addition to non-coated soot, Weingartner et al. (2003) determined $C_{ref}$ for coated particles as well, and the resulting $C_{ref}$ was higher, about 3.6. This is also closer to our observations, which is explained by the fact that at SMEAR II, the observed soot particles are likely aged and coated since there are no significant local emission sources. In these studies, however, the reference instruments were not filter-based photometers, and that can have an effect on the results.

Report 227 by WMO and GAW recommends determining $\sigma_{abs}$ from Aethalometer measurements by using a $C_{ref}$ value of 3.5 and not applying any filter-loading correction or particle-scattering reduction to the data. $C_{ref}$ was determined as an average over several data sets collected from different GAW stations. Comparing this value to $C_{NC}$, using the recommended $C_{ref} = 3.5$ would systematically underestimate $\sigma_{abs}$ at SMEAR II by $\sim 20\,\%$. It must also be noted that due to the lack of $R(ATN)$ correction, the BC concentration or $\sigma_{abs}$ may differ by as much as 50 % even if the true $\sigma_{abs}$ were to stay constant (Arnott et al., 2005). Therefore, in some cases, the data user may want to take the error caused by the filter loading into account and to use different correction algorithms, for example, when studying shorter time periods (e.g., a few days of data, which may fit a few filter spot changes causing apparent variation in the measured concentration).

There are both studies where a constant $C_{ref}$ has been used and studies where a wavelength-dependent $C_{ref}$ has been used. Others have observed no significant dependency of $C_{ref}$ on the wavelength (Backman et al., 2017; Bernardoni et al., 2021; Collaud Coen et al., 2010; Weingartner et al., 2003; WMO/GAW, 2016), and other studies have observed the opposite and shown that $C_{ref}$ is wavelength dependent

**Table 2.** Average values for the multiple-scattering correction factor ($C_{ref}$) for the different correction algorithms. The values are reported at 637 nm. The slope of the fit, standard error of the fit (SE), and 95 % confidence interval (CI) were determined by a linear regression applied to the whole data set. The median, mean, and standard deviation (SD), as well as the 5th and 95th percentile range, were determined from the $C_{ref}$ values that were calculated for each data point separately.

| | Fit | SE | 95 % CI | Median | Mean | SD | 5th and 95th percentiles |
|---|---|---|---|---|---|---|---|
| $C_{WEI}$ | 3.00 | 0.003 | [2.99, 3.00] | 3.34 | 3.29 | 0.57 | [2.59, 4.26] |
| $C_{ARN}$ | | | | 3.13 | 3.13 | 0.45 | [2.49, 3.81] |
| $C_{VIR}$ | 3.14 | 0.002 | [3.13, 3.14] | 3.30 | 3.28 | 0.56 | [2.53, 4.18] |
| $C_{COL}$ | 2.99 | 0.003 | [2.98, 2.99] | 3.28 | 3.32 | 0.57 | [2.55, 4.23] |
| $C_{NC}$ | 2.77 | 0.003 | [2.76, 2.77] | 3.09 | 3.06 | 0.55 | [2.32, 3.95] |

**Table 3.** Linear fits between the AE31 and reference absorption ($\sigma_{abs,ref}$ measured by the MAAP) for different ATN intervals (at 660 nm) as well as between the PSAP and $\sigma_{abs,ref}$. The value in parentheses is the coefficient of determination ($R^2$).

| ATN | 0–20 | 20–40 | 40–60 | 60–80 |
|---|---|---|---|---|
| W2003 | $1.05 \cdot x + 0.07$ (0.98) | $0.99 \cdot x + 0.15$ (0.97) | $0.95 \cdot x + 0.15$ (0.96) | $0.97 \cdot x + 0.12$ (0.94) |
| A2005 | $0.93 \cdot x + 0.06$ (0.96) | $0.93 \cdot x + 0.16$ (0.95) | $0.97 \cdot x + 0.15$ (0.92) | $1.06 \cdot x + 0.10$ (0.90) |
| V2007 | $1.02 \cdot x + 0.05$ (0.98) | $0.98 \cdot x + 0.10$ (0.98) | $0.97 \cdot x + 0.08$ (0.97) | $0.99 \cdot x + 0.05$ (0.96) |
| CC2010 | $1.01 \cdot x + 0.06$ (0.98) | $0.95 \cdot x + 0.15$ (0.97) | $0.92 \cdot x + 0.16$ (0.95) | $0.95 \cdot x + 0.13$ (0.92) |
| Non-corrected | $1.12 \cdot x + 0.07$ (0.98) | $1.01 \cdot x + 0.15$ (0.97) | $0.93 \cdot x + 0.14$ (0.96) | $0.93 \cdot x + 0.12$ (0.95) |
| Tr | 1–0.7 | 1–0.4 | 0.4–0.7 | 0–0.4 |
| B1999 | $1.04 \cdot x + 0.01$ (0.97) | $1.06 \cdot x + 0.02$ (0.97) | $1.06 \cdot x + 0.07$ (0.97) | $1.11 \cdot x + 0.02$ (0.98) |
| V2010 | $1.01 \cdot x - 0.02$ (0.96) | $1.12 \cdot x - 0.07$ (0.94) | $1.17 \cdot x + 0.01$ (0.95) | $1.46 \cdot x - 0.20$ (0.96) |

(Arnott et al., 2005; Kim et al., 2019; Schmid et al., 2006). These studies suggested that $C_{ref}$ increased with wavelength (i.e., filter fibers scattered more light at longer wavelengths). Interestingly, even though the wavelength dependency was not statistically significant, Weingartner et al. (2003) reported that the $C_{ref}$ obtained for internal mixtures of diesel soot and ammonium sulfate and coated Palas soot yielded $C_{ref} = 3.9 \cdot (\lambda/660\,\text{nm})^{0.18}$ and $C_{ref} = 3.66 \cdot (\lambda/660\,\text{nm})^{0.23}$, respectively, as can be calculated from their Table 3. The exponents are very close to the value of 0.18 obtained from the fittings to the Arnott et al. (2005) Table 1. Kim et al. (2019) found that $C_{ref}$ depended on wavelength even more strongly; a fitting to their Table 2 yielded $C_{ref} = 4.48(\lambda/532\,\text{nm})^{0.48}$. Because the results between the different studies vary, it is difficult to conclude whether $C_{ref}$ is wavelength dependent or not. To study the wavelength dependency of $C_{ref}$, it would be ideal to use a photoacoustic method or $\sigma_{ext}$-$\sigma_{sca}$ method for the reference measurements, since they are independent from the filter artifacts.

A newer model of the Aethalometer, AE33, applies the so-called dual-spot correction, so the instrument operators do not need to apply the correction algorithms themselves. However, the value of $C_{ref}$ is also an open question for the AE33, but since its filter material is different from the one used in the AE31, the results of the present study are not applicable to it. The filter material in AE33 is Teflon-coated

glass filter tape (Pallflex type T60A20), but the "old" filter tape (Q250F) has also been used with AE33, and the recommended $C_{ref}$ values to use with these filters are 1.57 and 2.14, respectively (Drinovec et al., 2015).

The different $C_{ref}$ values were not only determined as a linear fit that considered the whole time series. In addition to the results from linear fits, Table 2 presents the median, mean, and standard deviation of different $C_{ref}$ values that were determined separately for each data point according to Eq. (19). Determining $C_{ref}$ separately for each data point enabled studying the temporal variation in $C_{ref}$; for example, the times series of the different $C_{ref}$ values are presented in Fig. S2 in the Supplement. The median and mean values differed somewhat from the slopes of the linear fits, which were about 10 % lower than the median values. Comparing the median and mean values shows no large difference, meaning that the $C_{ref}$ values were rather normally distributed. The variation in median $C_{ref}$ values between the different correction algorithms was small compared to the relatively large standard deviation (see Table 2).

$C_{ref}$, determined separately for each data point, was not stable over time (see time series presented in Fig. S2 in the Supplement), and we observed seasonal variation for $C_{ref}$, which is presented in Fig. 2 for $C_{NC}$ as an example. The seasonal variation was observed not only for $C_{NC}$ but also for $C_{WEI}$ and $C_{COL}$, presented in Fig. S3 in the Supplement. Fig-

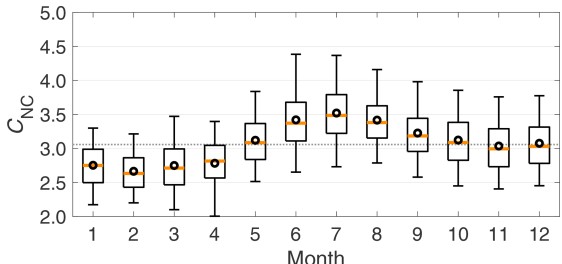

**Figure 2.** The seasonal variation in the multiple-scattering correction factor for non-corrected data ($C_{NC}$). The orange line in the middle of the box is the median, the black circle is the mean, the edges of the boxes represent the 25th and 75th percentiles, and the whiskers represent the 10th and 90th percentiles of the data. The dashed line is the median for all data.

ure 2 shows that $C_{NC}$ was clearly above the median during the summer and below the median in winter and early spring. $C_{NC}$ reached its maxima in July and its minima in February. For $C_{VIR}$ and $C_{ARN}$, the seasonal variation was much less
pronounced (Fig. S3 in the Supplement).

Since $C_{WEI}$ and $C_{COL}$ had similar seasonal variation, it is unlikely that the seasonal variation observed for $C_{NC}$ was caused by the lack of filter-loading correction. There was seasonal variation for $C_{COL}$ as well, and for example, the sea-
10 sonal variations between $C_{WEI}$ and $C_{COL}$ were rather similar, even though we applied constant $f$ values in W2003. The CC2010 algorithm considers the wavelength dependency of scattering and the $\omega$ of the accumulated particles. It is rather surprising that taking these parameters, which have seasonal
variation at SMEAR II (Luoma et al., 2019; Virkkula et al., 2011), into account did not seem to reduce the seasonality of $C_{ref}$.

The seasonal variations in $C_{ARN}$ and $C_{VIR}$ were less obvious than in $C_{WEI}$, $C_{COL}$, and $C_{NC}$. The lower seasonal vari-
20 ation for $C_{ARN}$ might be explained by the subtraction of the scattering fraction before the loading correction was applied and $C_{ARN}$ was determined. The fact that $C_{ARN}$ has fewer data points than the other $C_{ref}$ values might also explain part of the lower seasonality. For $C_{VIR}$, the lack of seasonal varia-
25 tion was probably caused by the very strong seasonal variation in the compensation parameter ($k$; see Fig. 9a) as will be discussed below in Sect. 4.4. The V2007 algorithm does not assume any coefficients but depends only on the difference between the last and first measurements of the filter spots.
Therefore, it seems to adjust to seasonal changes, whereas the other algorithms apply coefficients. According to our results, V2007 and A2005 accounted for the variations in the optical properties of the particles embedded in the filter well, and therefore the seasonal variations in $C_{VIR}$ and $C_{ARN}$ were
reduced.

As indicated by the seasonal variation, $C_{ref}$ was not a constant value, but it depended on the optical properties of the particles embedded in the filter. As stated before, Weingart-

ner et al. (2003) and Arnott et al. (2005) observed different $C_{ref}$ values for different types of aerosols, so $C_{ref}$ was lower
for "pure" soot (no coating) and higher for coated soot or ambient aerosol particles. This suggests that $C_{ref}$ increases with increasing $\omega$. This supports our observations, since at SMEAR II, $\omega$ is the highest in summer and lowest in winter (Luoma et al., 2019; Virkkula et al., 2011). However, Collaud
Coen et al. (2010) observed a decreasing trend for $C_{ref}$ with increasing $\omega$ when they compared the average conditions at several stations.

$\omega$, however, is not the only optical property of aerosol particles that had a clear seasonal variation (Luoma et al., 2019;
Virkkula et al., 2011). For example, the size-dependent $b$ and $\alpha_{sca}$ reached their maxima in summer and minima in winter, which indicated that in summer the fraction of smaller particles increased. Luoma et al. (2019) showed that the seasonal variation in $b$ and $\alpha_{sca}$ is explained especially by the differ-
ences in the accumulation mode (particles in the size range of 100 nm–1 μm) particle concentration and size distribution: in summer, the volume concentration peaks at around 250 nm and in winter at around 350 nm. The size distribution affects the penetration depth of the particles as smaller particles pen-
etrate deeper in the filter (Moteki et al., 2010). Scattering particles that penetrate deeper in the filter increase the multiple scattering in the filter, and that could be one explanation for higher $C_{ref}$ values observed in summer.

The differences in the scattering properties of differently
sized particles might also explain the observed seasonal variation in $C_{ref}$. The correction algorithms only consider the amount of scattering and not the direction of scattering. Smaller particles scatter relatively more light in the backward direction, which increases the optical path of the light
ray through the filter (i.e., $C_{ref}$ should increase). Therefore, this effect may cause the observed increase in the multiple-scattering correction factor $C_{ref}$ in summer. This could also explain why $C_{VIR}$ had no seasonal dependency; the compensation parameter seemed to also depend on $b$ (see Sect. 4.4),
and that would make V2007 the only algorithm that takes the direction of the particulate scattering into account. Note that V2007 does not take $b$ into account directly, but it seems to influence the calculated compensation parameter (see Sect. 4.4).

However, only very weak correlation was found between $C_{NC}$ and $\omega$ ($R = 0.17$; $p$ value $< 0.05$) and $C_{NC}$ and $b$ ($R = 0.23$; $p$ value $< 0.05$), so $\omega$ and $b$ do not necessarily explain the observed seasonal variations in the $C_{ref}$ values. For $C_{WEI}$
and $C_{COL}$, the results were similar, but for $C_{VIR}$, the $R$ values were even lower and even insignificant for $\omega$.

We observed slightly higher correlation ($R = 0.30$; $p$ value $< 0.05$) between $C_{NC}$ and relative humidity (RH), which is presented in Fig. 3 (the correlation was similar for $C_{WEI}$ and $C_{COL}$ but weaker, about 0.09, for $C_{VIR}$). There-
90 fore, one possible reason for the observed seasonal variation in the different $C_{ref}$ values could be caused by changes in the instrumental RH and the RH differences between the MAAP

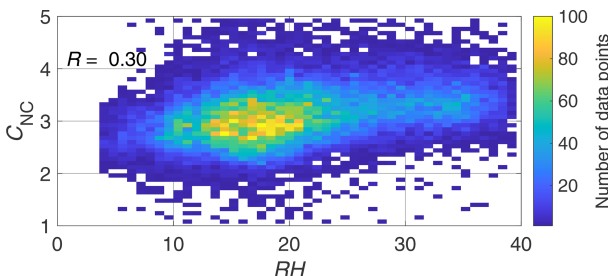

**Figure 3.** The dependency of the multiple-scattering correction factor for non-corrected data ($C_{NC}$) on the instrumental relative humidity (RH) in the MAAP. The colored grid points represent the number of data points in each grid point. There are 50 grid points in the $x$ and $y$ directions, so in total there are 2500 grid points.

and AE31. The RH presented in Fig. 3 was measured in the MAAP, and it varied between 5 %–40 % since the periods when the filter of the MAAP was exposed for RH equal to or larger than 40 % were excluded from this study. Because the AE31 was equipped with Nafion dryers, the RH in the AE31 varied less and was in the range of 5 %–20 %. The RH can influence filter-based optical measurements by affecting the optical properties of the aerosol particles and the filter fibers as well as by affecting the penetration depth of particles in the filter medium. The effect of the rate of change in RH on $C_{ref}$ was also studied, but the rate of change in RH did not show any correlation with $C_{NC}$.

Due to hygroscopic growth, the aerosol particles scatter more light in humid conditions compared to dry conditions. The enhanced scattering induced by higher RH could then increase the scattering and optical path in a particle-laden filter medium. However, at SMEAR II, increasing RH should have caused a decrease in $C_{NC}$, since hygroscopic growth would have increased the particulate scattering especially in the reference instrument MAAP. Hygroscopic growth may also affect the penetration depth of the particles in the filter (Moteki et al., 2010). When particles penetrate deeper in the filter, the effect of the multiple scattering is higher, increasing the measured $\sigma_{ATN}$. Because the RH in the MAAP was higher than in the AE31, the particles directed in the AE31 may have penetrated relatively deeper in the filter than the particles directed in the MAAP filter, in summer, larger difference in the RH between the instruments could have increased the measured $C_{ref}$. However, hygroscopic growth should not be significant in RH conditions below 40 %, which is why the effects related to hygroscopic growth seem unlikely explanations.

Also, the optical properties of the filter may change if the filter is exposed to high-RH conditions. The aerosol particles may take up water even below supersaturation, and when liquid particles collide on the filter the moisture is taken up by the filter. Kanaya et al. (2013) compared a MAAP against a continuous soot monitoring system (COSMOS; Miyzaki et al., 2008) and actually observed a slight dependency in the $\sigma_{abs}$ measured by the MAAP, so at low RH ($< 40$ %) $\sigma_{abs}$ increased with increasing RH, which is contrary to our results as we observed that the MAAP observed relatively lower $\sigma_{abs}$ at higher RH. However, the authors also observed the opposite behavior at higher RH ($> 50$ %). They suggested that the RH affected the surface roughness of the filter, which is used in the radiative transfer scheme (Petzold and Schönlinner, 2004), and therefore could have affected $C_{ref}$.

The results showed that even though we excluded the high-RH data, the instruments seemed to be sensitive to variations in RH even below the recommended 40 %. However, the reason for the sensitivity remains unclear and would require more research and measurements, and therefore further analysis is omitted from the scope of this article.

## 4.2 Performance of the correction algorithms

In this section, we included data from June 2013 to February 2016 to have all three absorption instruments running in parallel to prevent any differences caused by different periods.

Since the $\sigma_{abs}$ derived from AE31 measurements used the $C_{ref}$ values determined here, the $\sigma_{abs}$ measurements of the AE31 and MAAP were expected to agree well, which is shown in Fig. 4. The AE31 data in Fig. 4 were produced by applying the $C_{ref}$ values determined from the linear fits (Table 2, column "Fit"). The correlation coefficients and slopes of the linear fits presented in Fig. 4 were close to unity. The AE31 correction schemes underestimated $\sigma_{abs}$ only slightly, and the slopes varied from 0.96 to 1.00. The AE31 data corrected with A2005 and CC2010 underestimated $\sigma_{abs}$ the most (slopes of the linear fits were 0.97 and 0.96, respectively). The reduction in particulate scattering in CC2010 after applying the multiple-scattering correction (i.e., $C_{ref}$) could explain the slight underestimation in CC2010-derived data. For the underestimation in A2005-derived data, the reason is probably the different way of determining $C_{ARN}$ compared to other $C_{ref}$ values. The iterative manner of determining $C_{ARN}$ separately for each filter spot and then taking the median from these values was not as successful as the linear-fit method, which was used for the other algorithms. However, the underestimation for A2005 and CC2010 are only minor.

Surprisingly, the non-corrected (NC) AE31 data (Fig. 4e) did not seem to have a significant difference in the correlation coefficient compared to, for example, the data corrected with W2003 or CC2010 (Fig. 4a and d, respectively). However, the relation between $\sigma_{abs,NC}$ and $\sigma_{ref}$ depended more on ATN than it did for any filter-loading-corrected data, which is shown by the color coding (ATN) of the data points and in Table 3, which presents the slopes of the linear fits and $R^2$ values for different ATN intervals. If only data from a highly loaded filter (ATN $> 60$ at 660 nm) were taken into account, the slopes of the linear fits were 0.97, 1.06, 0.99, 0.95, and 0.93 for W2003, A2005, V2007, CC2010, and NC, respectively. The smallest decrease in the slope with increasing ATN determined for the loaded filter was observed for

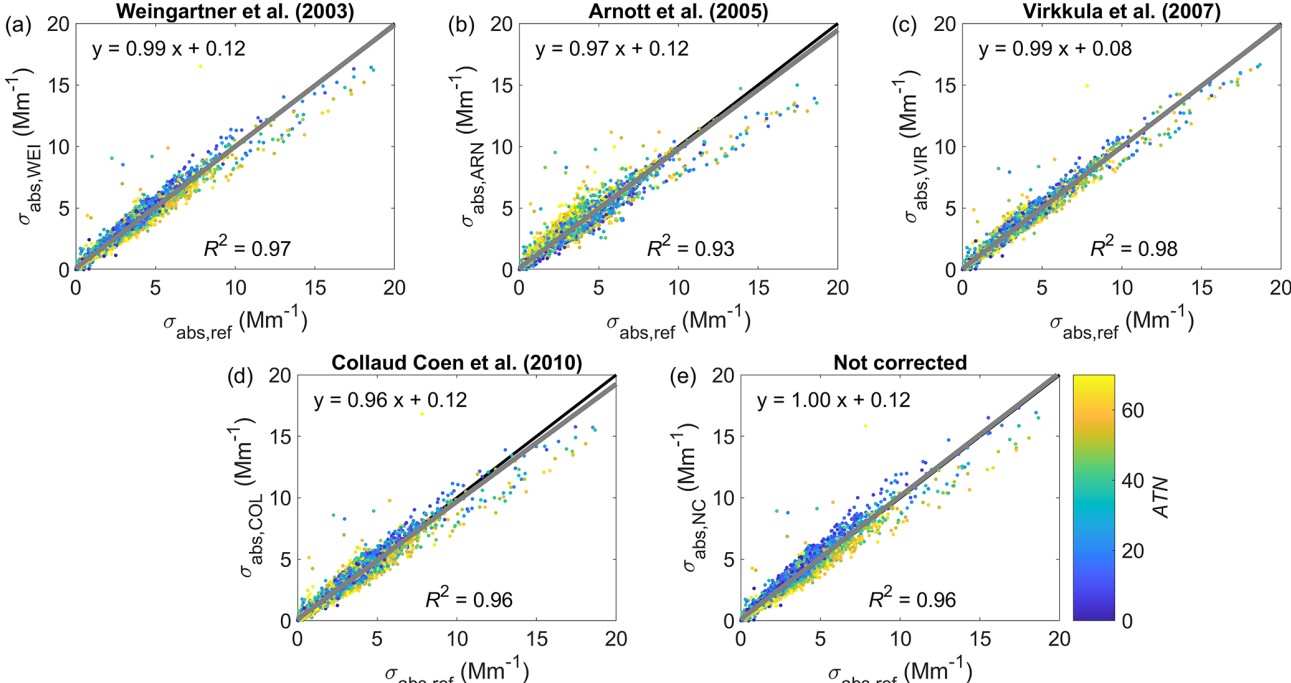

**Figure 4.** Comparison of the AE31 and MAAP measurements for all the different AE31 correction algorithms. The corrected AE31 data have been interpolated to the same wavelength as that of the MAAP (637 nm). The data points are colored by the AE31 filter attenuation (ATN; at 660 nm). The fit to the data is presented with a grey line, and the equation and the coefficient of determination ($R^2$) are shown in the panels. The 1 : 1 line is shown in black.

data that were corrected by V2007. Interestingly, the slopes for the loaded filter increased for data that were corrected by A2005. This different behavior is probably caused by the fact that the A2005 algorithm did not consider the loading through ATN but applied a cumulative $\sigma_{\mathrm{abs}}$, which apparently at SMEAR II seemed to overestimate the loading and loading correction, thus leading to an increasing slope with ATN. The biggest decrease in the slope determined for a highly loaded filter was observed for the NC data, as expected.

According to the $R^2$ values presented in Table 3, the precision of the AE31 decreased with increasing ATN. For example, for the data corrected with the A2005 algorithm, $R^2$ decreased from 0.96 for a clean filter (ATN $< 20$) to 0.90 for a loaded filter (ATN $> 60$). However, the decrease in $R^2$ was quite minor. Miyakawa et al. (2020) also observed rather high $R^2$ values between an Aethalometer (model AE51) and a reference instrument (single-particle soot photometer and COSMOS) when ATN was below 70, but when ATN exceeded 70, $R^2$ decreased more rapidly. Unlike for the AE31, the loading on the filter did not seem to affect the precision of the PSAP at all as the $R^2$ values did not decrease with increasing loading.

As presented in Table 3, the linear fits for the AE31 and PSAP data against the reference did not have an intercept of zero. This could be caused by the scattering artifact and the fact that the correction algorithms failed to take the scatter-

ing artifact partly into account. The intercept is the smallest for the B1999-corrected PSAP data and the largest for AE31 data. A fraction of $\sigma_{\mathrm{sca}}$ is subtracted in the AE31 algorithms by A2005 and CC2010. However, the data corrected with these algorithms still have a higher intercept than or similar intercept to the non-corrected data and the data corrected by the W2003 and V2007 algorithms. Considering the intercept, the V2007-corrected data perform the best in the AE31-vs.-MAAP comparison, which is slightly surprising, since it does not take the scattering subtraction into account. For the V2010-corrected PSAP data, the intercept is negative, suggesting that the V2010 algorithm overestimates the apparent absorption by scattering particles.

The comparison between the MAAP and the PSAP is presented in Fig. 5a and b and in Table 3 for both the correction schemes B1999 and V2010, respectively. Figure 5b shows that V2010 overestimated $\sigma_{\mathrm{abs}}$ especially when the loading was high (Tr was low), and the linear regression was 1.25. B1999 also overestimated $\sigma_{\mathrm{abs}}$ slightly, but in general it performed better in comparison with the MAAP, and the slope was 1.07 (Fig. 5a). The linear fits in Fig. 5a and b include all the data, but Table 3 presents the slopes of the linear fits for data with different Tr limits. It is actually recommended to use PSAP data with Tr $> 0.7$, and if only these data are taken into account, especially the data corrected with the V2010 algorithm, they perform much better and have a slope of 1.01,

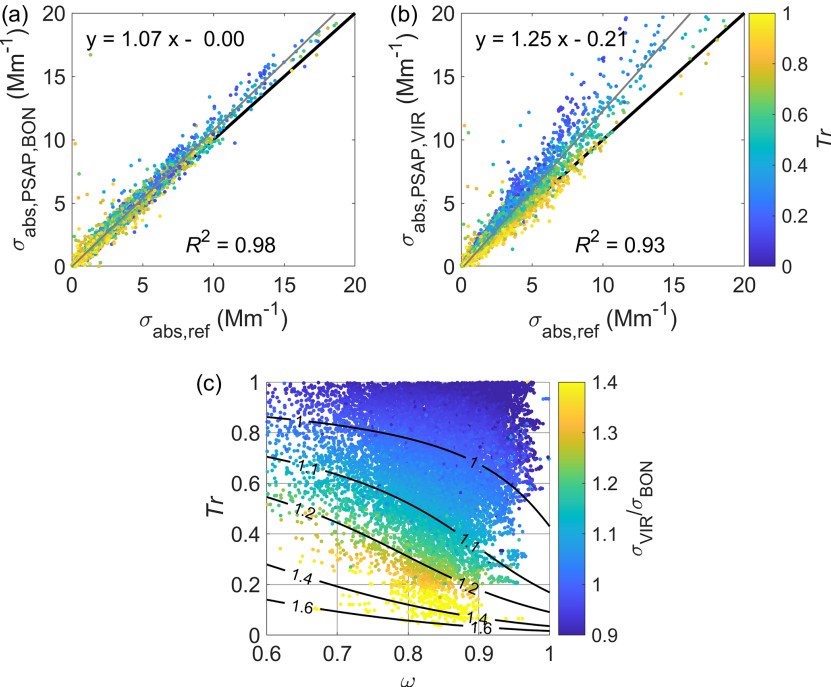

**Figure 5.** Panels (**a** and **b**) present the comparison of the PSAP and MAAP measurements for the B1999 and V2010 correction algorithms, respectively. The data points are colored by the PSAP filter transmittance (Tr); the fit to the data is presented with a grey line, and the equation and the coefficient of determination ($R^2$) are shown in the panels. The 1 : 1 line is shown in black. Panel (**c**) presents the relation of the PSAP-derived absorption coefficients corrected with the V2010 algorithm ($\sigma_{abs,PSAP,VIR}$), and the B1999 algorithm depends on Tr and the single-scattering albedo ($\omega$). The contour lines show the theoretically determined $\sigma_{abs,PSAP,VIR}/\sigma_{abs,PSAP,BON}$ ratio. $\omega$ was determined from nephelometer and MAAP measurements at 637 nm.

but the data derived with the B1999 algorithm also yield a smaller slope of 1.04.

If all the data were included in the comparison, as in Fig. 5a and b, the overestimation of $\sigma_{abs}$ would suggest also deriving the $C_{ref}$ values for the PSAP data. Here, we did not derive the $C_{ref}$ values for the PSAP since they are not typically used in a similar way to deriving $\sigma_{abs}$ from the AE31 measurements. In general, the multiple scattering does not cause such a big artifact in filter material typically used in the PSAP compared to in the thicker AE31 filters. However, if we considered only the data where Tr < 0.7, the PSAP and MAAP agree well for both correction algorithms. This result then suggests that there is no need for deriving a new $C_{ref}$ for the PSAP. Svensson et al. (2019) studied the multiple scattering in quartz filters, and they derived the equations that can be used in determining the $C_{ref}$ value for the PSAP. Differently to AE31 correction algorithms, the $C_{ref}$ used in PSAP algorithms is included in the coefficients of Eqs. (13)–(15), and therefore determining $C_{ref}$ for the PSAP is not as straightforward.

The differences between these two correction algorithms are studied in more detail in Fig. 5c, which shows how the algorithms perform with different Tr and $\omega$ values. As discussed before, V2010 produces notably higher $\sigma_{abs}$ values when the filter is highly loaded (Tr < 0.5). However, the dif-

ference between the algorithms depends not only on Tr but also on $\omega$, so at high $\omega$ and Tr, $\sigma_{abs,PSAP,VIR}/\sigma_{abs,PSAP,BON}$ < 1, and when $\omega$ decreases the $\sigma_{abs,PSAP,VIR}/\sigma_{abs,PSAP,BON}$ ratio grows. The reason for this is that the V2010 algorithm is a function of $\omega$.

The dependency of $\sigma_{abs}$ on ATN and Tr is presented in the Supplement (Figs. S4 and S5). On average, the decrease in $\sigma_{abs}$, which was not corrected for the filter loading ($\sigma_{abs,NC}$ and $\sigma_{abs,PSAP,ATN}$), with the increasing ATN and decreasing Tr was not clear. This effect is better seen in the results presented in Table 3. However, Fig. S5 in the Supplement shows that especially for the PSAP, the use of correction algorithms decreased the variation, which is a strong recommendation for using the correction algorithms. This is also seen in the AE31 data, but the effect was less notable (Fig. S4 in the Supplement).

Because it is impossible to separate the effect of different size cuts from a loaded filter, here the $PM_1$ and $PM_{10}$ measurements were combined and averaged together. In general, $PM_1$ accounted for about 90 % of the $PM_{10}$ $\sigma_{abs}$; for $\sigma_{sca}$ the fraction of $PM_1$ was about 75 % (Luoma et al., 2019). Because absorbing particles, which are considered to consist mostly of black carbon, are typically in the fine mode (diameter < 1 µm), $\sigma_{abs}$ is not expected to deviate much between the different size cuts. However, the differing size cuts,

which cause more deviation in $\sigma_{sca}$, could have affected the $\sigma_{abs}$ measurements since the particulate scattering causes apparent absorption and affects the multiple scattering in the filter. For example, the coarse particles (diameter > 1 μm) do not penetrate as deep in the filter as the fine-mode particles, which could possibly influence the $C_{ref}$ values. In an ideal situation the PM$_1$ and PM$_{10}$ absorption would have been measured by separate instruments.

Our observations underline the need for filter-loading correction, especially if one studies shorter time periods. For longer time periods (e.g., trend analysis or studies of seasonal variation), the effect of ATN on the variation smooths out, but for shorter time periods (e.g., case studies), the changing ATN can have a notable effect on the results if no filter-loading correction is applied. However, when not correcting for the filter-loading effect, the precision of the instrument and $\sigma_{abs}$ or the BC concentration on average are reduced, which is why applying a filter-loading correction on filter-based photometers is always recommended.

## 4.3 Absorption Ångström exponent for different correction algorithms

The effect of the correction algorithms on $\alpha_{abs}$ was studied, and the average $\alpha_{abs}$ values for different correction algorithms of the AE31 and PSAP are presented in Fig. 6. This figure includes only parallel data from both the AE31 and the PSAP in order to avoid any differences caused by different time periods. For a comparison, $\alpha_{abs}$ was also determined for the "raw" PSAP data that were not corrected by any algorithms (i.e., $\sigma_{ATN}$; see Eq. 1). To have comparable $\alpha_{abs}$ values from the different instruments, Fig. 6 includes only overlapping AE31 and PSAP data from 2011–2015. Since the PSAP operates at three wavelengths (467, 530, and 660 nm), we determined the AE31-related $\alpha_{abs}$ in Fig. 6 by using only the wavelengths 470, 520, 590, and 660 nm of the AE31. The rest of the AE31 wavelengths were omitted from this comparison to minimize the effect of different wavelength ranges have on $\alpha_{abs}$ (for example, see Luoma et al., 2019; Table 1). $\alpha_{abs}$ was determined as a linear fit over all the selected wavelengths according to Eq. (16). Since Luoma et al. (2019) did not observe a big difference between the PM$_1$ and PM$_{10}$ $\alpha_{abs}$, we included both measurements in this comparison.

According to Fig. 6, the median values of $\alpha_{abs}$ varied notably between the different instruments and correction algorithms: the lowest median value of $\alpha_{abs}$ was 0.93, and it was measured by the AE31 and corrected by CC2010; and the highest median value of $\alpha_{abs}$ was 1.54, and it was measured by the PSAP and corrected by V2010. The difference between the highest and lowest median values of $\alpha_{abs}$ was about 1.7-fold. The correction algorithms were applied to each wavelength separately, and therefore the correction algorithms affected the wavelength dependency of the derived $\sigma_{abs}$. The scattering and loading corrections are different for each wavelength because for example $\sigma_{sca}$, $\omega$, ATN,

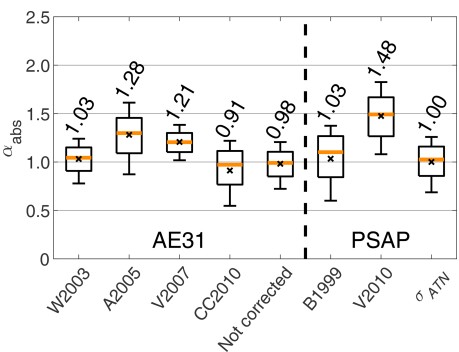

**Figure 6.** The absorption Ångström exponent ($\alpha_{abs}$) for all the different AE31 and PSAP correction algorithms. The orange line in the middle of the box is the median; the black circle is the mean; the edges of the boxes represent the 25th and 75th percentiles, and the whiskers represent the 10th and 90th percentiles of the data. The values given above each box show the corresponding median values.

and Tr, which are used in the algorithms, are wavelength dependent. For the AE31, we studied the same five correction algorithms as in Sect. 4.1. The lowest median $\alpha_{abs}$ values were observed for the non-corrected data ($\alpha_{abs,AE,NC}$) and for data that were corrected with the CC2010 and W2003 algorithms ($\alpha_{abs,AE,COL}$ and $\alpha_{abs,AE,WEI}$). The median $\alpha_{abs}$ values for the data corrected with the A2005 and V2007 algorithms ($\alpha_{abs,AE,ARN}$ and $\alpha_{abs,AE,VIR}$) were higher at 1.20 and 1.19, respectively.

A2005 was the only algorithm that assumed a wavelength-dependent $C_{ref}$. Since $C_{ARN}$ increased with wavelength (i.e., bigger correction due to multiple scattering at higher wavelengths), taking the wavelength dependency of $C_{ref}$ into account increases $\alpha_{abs,AE,ARN}$ compared to other algorithms. The correction factor of the V2007 algorithm depended on the difference between the ATN of loaded and clean filter spots. Most of the time ATN increased faster at short wavelengths than at long wavelengths, so the difference between the ATN of the loaded and clean filter spots was higher than for longer wavelengths. Therefore, the filter-loading correction was bigger for shorter wavelengths, and after the correction the difference between the $\sigma_{abs}$ values at different wavelengths increased, increasing $\alpha_{abs}$ as well.

For the PSAP data, the $\alpha_{abs}$ values were generally a little higher compared to the AE31-derived $\alpha_{abs}$. The lowest PSAP-derived median value for $\alpha_{abs,PSAP,NC}$ was 1.01, which resulted from data that were not corrected by any algorithm. B1999 resulted in a median $\alpha_{abs,PSAP,BON}$ value of 1.04, and V2010 produced the highest $\alpha_{abs,PSAP,VIR}$ overall, which was 1.48. A similar order of the average $\alpha_{abs}$ values from different algorithms was observed by Backman et al. (2014) at an urban station in Elandsfontein, South Africa. For a data set measured off the east coast of the United States on a research ship, Backman et al. (2014) also reported the highest $\alpha_{abs}$ for V2010. These results are consistent with

each other. The explanation is that in V2010 all constants are wavelength dependent, contrary to in B1999.

The differences between the correction algorithms could possible be decreased by adding or reducing the wavelength dependency of the constant values used. Since we did not have reference measurements at several wavelengths, it is impossible to say which one of the correction algorithms yielded the most truthful value for $\alpha_{abs}$. This could be determined with several MAAPs operating at different wavelengths by measuring the particles suspended in the air by the photoacoustic method (Kim et al., 2019), by a polar photometer (Bernardoni et al., 2021), or by a multi-wavelength absorption analyzer (MWAA; Massabò et al., 2013). According to the comparison between an AE31 and an MWAA by Saturno et al. (2017), the best agreement for $\alpha_{abs}$ was achieved with uncorrected AE31 data, and the AE31 data corrected by CC2010 also agreed well with the reference measurements.

We also studied if the $\alpha_{abs}$ values were affected as the filter became more loaded with particles. Figure 7 presents the $\alpha_{abs}$ values derived from AE31 data corrected with different algorithms as a function of ATN, and Fig. 8 presents the $\alpha_{abs}$ values derived from PSAP data as a function of Tr. The $\alpha_{abs}$ derived from corrected PSAP data (Fig. 8a and b) did not seem to depend on loading at Tr > 0.4, but for higher filter loadings $\alpha_{abs}$ still increased with decreasing Tr with both B1999 and V2010 corrections. In comparison, for the non-corrected PSAP data (i.e., $\sigma_{ATN}$), $\alpha_{abs}$ decreased with increasing Tr (Fig. 8c). The $\alpha_{abs}$ values derived from the AE31 data were also studied; $\alpha_{abs,AE,WEI}$, $\alpha_{abs,AE,ARN}$, $\alpha_{abs,AE,COL}$, and $\alpha_{abs,AE,NC}$ clearly decreased with increasing ATN. If ATN increased from 5 to 70, the decreases in $\alpha_{abs,AE,WEI}$, $\alpha_{abs,AE,ARN}$, $\alpha_{abs,AE,COL}$, and $\alpha_{abs,AE,NC}$ were rather linear and around $-22\%$, $-23\%$, $-33\%$, and $-27\%$, respectively.

The $\alpha_{abs,AE,VIR}$ derived from data corrected with V2007 did not seem to depend on ATN if not taking into account very high filter loadings (ATN at 660 nm > 70; on average the filter changed when ATN at 660 nm $\approx$ 90). In V2007, $k$ was determined for each wavelength separately. $k$ is often larger for the shorter wavelengths, which means that the non-linearity caused by the increased filter loading is relatively stronger at the shorter wavelengths (Drinovec et al., 2017; Virkkula et al., 2007, 2015), which was also observed by this study (discussed in the next section). According to these results, the algorithms other than V2007 do not seem to account for the wavelength dependency of $R(ATN)$ enough.

## 4.4 Variations in the compensation parameter

The variation in $k$ at SMEAR II has already been studied by Virkkula et al. (2007), who used AE31 data from December 2004 to September 2006. During this period, the AE31 was operating without any cutoff and there were no scattering measurements available, and this period was not included

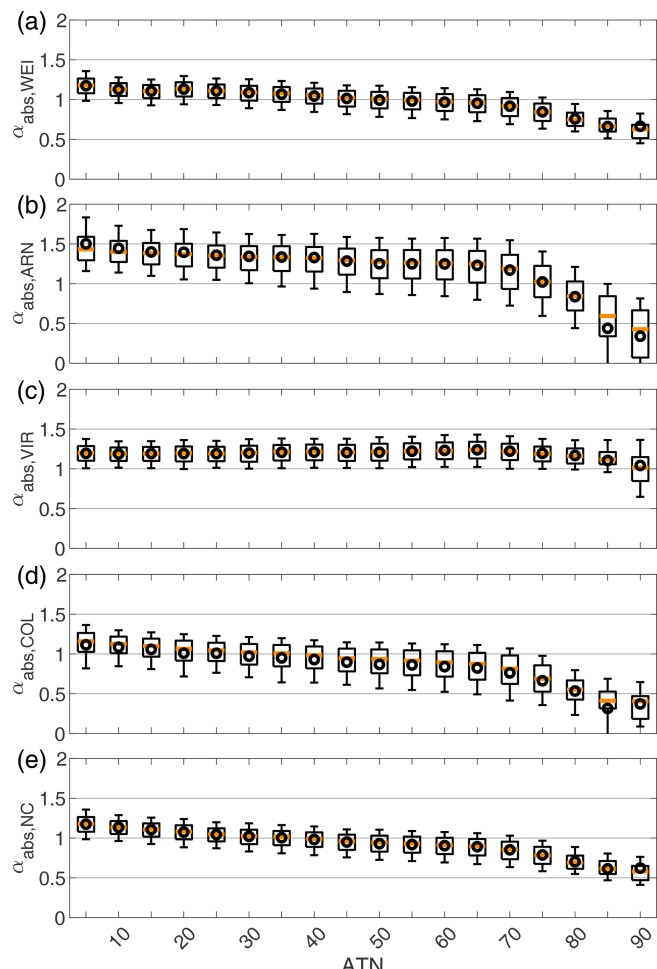

**Figure 7.** The dependency of the absorption Ångström exponent ($\alpha_{abs}$) on the AE31 filter attenuation (ATN; at 660 nm) for different correction algorithms. The orange line in the middle of the box is the monthly median; the black circle is the mean; the edges of the boxes represent the 25th and 75th percentiles, and the whiskers represent the 10th and 90th percentiles of the data.

in our study. Here, we repeated the analysis for a longer time series and included the $\sigma_{sca}$ measurements, so we could also determine $b$ and $\omega$.

The average values of $k$ are presented in Table 4. The mean values of $k$ varied from $4.6 \times 10^{-3}$ at 370 nm to $2.0 \times 10^{-3}$ at 950 nm. The wavelength dependency of $k$ is described by $a_k$, which is the slope of a linear fit of $k$ over different wavelengths ($k_\lambda = a_k\lambda + k_0$; see example in Fig. 9b). A negative $a_k$ means that on average the filter-loading correction was greater at shorter wavelengths. The light attenuation is stronger at shorter wavelengths due to higher absorption and scattering by the particles, and therefore the shorter wavelengths are prone to bigger error caused by the filter loading. At longer wavelengths, the standard deviation of $k$ was higher, meaning that $k$ was more sensitive to the particle

**Table 4.** The mean compensation parameters ($k$) and the wavelength dependency of $k$ ($a_k$) for the AE31 correction algorithm suggested by Virkkula et al. (2007). The average values are calculated over all the seasons but also separately for each season. The seasons were classified as spring (March–May), summer (June–August), autumn (September–November), and winter (December–February).

| Season | 370 nm ($\times 10^{-3}$) | 470 nm ($\times 10^{-3}$) | 520 nm ($\times 10^{-3}$) | 590 nm ($\times 10^{-3}$) | 660 nm ($\times 10^{-3}$) | 880 nm ($\times 10^{-3}$) | 950 nm ($\times 10^{-3}$) | $a_k$ ($\times 10^{-6}$ nm$^{-1}$) |
|---|---|---|---|---|---|---|---|---|
| all | $4.6 \pm 7.0$ | $3.6 \pm 7.2$ | $3.5 \pm 8.0$ | $3.4 \pm 8.9$ | $2.7 \pm 9.5$ | $2.1 \pm 10.6$ | $2.0 \pm 10.8$ | $-4.2$ |
| Spring | $4.4 \pm 7.0$ | $3.5 \pm 6.5$ | $3.4 \pm 7.6$ | $3.4 \pm 8.8$ | $2.8 \pm 9.2$ | $2.2 \pm 10.5$ | $2.2 \pm 10.2$ | $-3.5$ |
| Summer | $3.0 \pm 6.7$ | $1.5 \pm 6.3$ | $1.1 \pm 6.9$ | $0.5 \pm 9.0$ | $-0.4 \pm 9.6$ | $-1.7 \pm 10.3$ | $-2.5 \pm 11.3$ | $-8.8$ |
| Autumn | $4.6 \pm 7.7$ | $3.6 \pm 9.1$ | $3.6 \pm 9.7$ | $3.5 \pm 10.3$ | $2.8 \pm 10.2$ | $2.5 \pm 10.9$ | $2.2 \pm 12.1$ | $-3.6$ |
| Winter | $5.5 \pm 6.3$ | $4.8 \pm 6.4$ | $4.8 \pm 7.0$ | $4.8 \pm 6.9$ | $4.3 \pm 8.4$ | $4.4 \pm 9.9$ | $4.3 \pm 9.1$ | $-1.7$ |

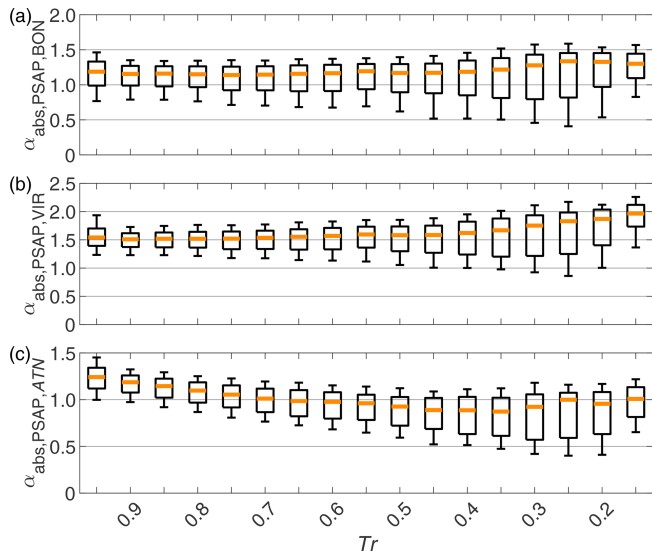

**Figure 8.** The dependency of the absorption Ångström exponent ($\alpha_{abs}$) on the PSAP filter transmittance (Tr) for **(a)** B1999, **(b)** V2010, and **(c)** non-corrected $\sigma_{ATN}$. The explanation for the box-plots is the same as in Fig. 5.

properties at longer wavelengths. The same observation was noted by Virkkula et al. (2015) as well.

At SMEAR II, we observed that $k$ and $a_k$ had a very strong seasonal variation, so $k$ and $a_k$ were the lowest in summer, which was also noted by Virkkula et al. (2007). The seasonal variation was observed at all wavelengths, but the variation was more pronounced at longer wavelengths. The seasonally averaged $k$ and $a_k$ are presented in Table 4, and an example of the seasonal variation in $k$ at 880 nm is presented in Fig. 9a. Similar seasonal patterns for $k$ were also observed by Virkkula et al. (2007), Wang et al. (2011), and Song et al. (2013). In summer, the mean $k$ values at the longer wavelengths (660–950 nm) were negative, meaning that without the correction, the AE31 would actually overestimate $\sigma_{abs}$ at longer wavelengths.

Previous studies (e.g., Virkkula et al., 2007; Wang et al., 2011; Song et al., 2013) have suggested that the seasonal variation in $k$ could be due to variations in $\omega$, with lower

$\omega$ inducing higher $k$. This behavior is observed at SMEAR II as shown in Fig. 9a and d; $\omega$ peaks in summer as $k$ has its minima, and the correlation coefficient between $\omega$ and $k$ is $-0.47$. The variation in $\omega$ also explains the observed negative $k$ values. Virkkula et al. (2007) stated that the negative values are associated with the response of the Aethalometer to scattering aerosols as the negative $k$ values are observed when $\omega$ is high. The effect of $\omega$ was taken into account, for example in the AE31 correction algorithms suggested by Weingartner et al. (2003), Arnott et al. (2005), and Collaud Coen et al. (2010). Virkkula et al. (2015) presented a theoretical explanation of the $\omega$ dependency of $k$, which our analysis supports.

Also, the effect being caused by the sizes of the particles has been suggested. The sizes of the particles affect their scattering properties and also their penetration depth in the filter that again could affect $k$. The size distribution of the particle population is described by $b$, with higher $b$ indicating smaller particles. Müller et al. (2014), for example, showed that the effect of the asymmetry parameter, which is a function of $b$ (Andrews et al., 2006), had an effect on the PSAP data.

The dependency of $k$ on both $b$ and $\omega$ was investigated more closely by Virkkula et al. (2015) at SORPES, an urban station located in Nanjing, China. The study showed positive correlation between $k$ and $b$ and negative correlation between $k$ and $\omega$. At SMEAR II, we also observed negative correlation between $k$ and $\omega$ (Fig. 9c). However, contrary to the results by Virkkula et al. (2015), we observed negative correlation between $k$ and $b$ (Fig. 9d). Virkkula et al. (2015) discussed difficulties of showing whether $b$ or $\omega$ was the dominant property in determining $k$. At SMEAR II, $\omega$ varies in a wider range compared to the observations at SORPES, which could explain some of the observed differences. The mean and standard deviation of $\omega$ at SMEAR II were $0.87 \pm 0.07$ (at 550 nm; Luoma et al., 2019) and at SORPES $0.93 \pm 0.03$ (at 520 nm; Shen et al., 2018). However, a clear reason for the negative correlation between $k$ and $b$ at SMEAR II was not found.

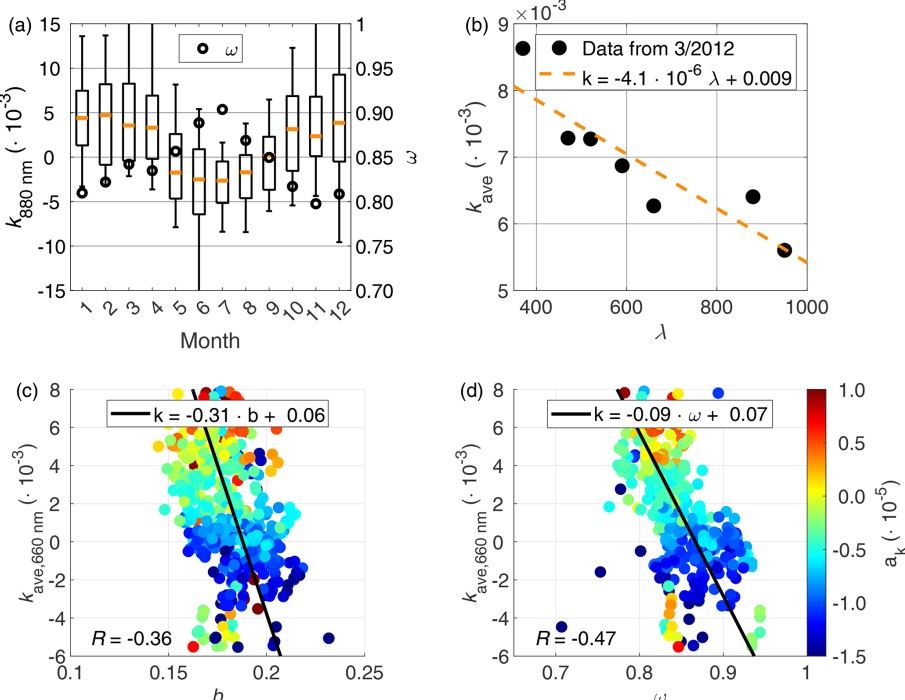

Figure 9. (a) The seasonal variation in the compensation parameter ($k$). (b) An example of calculating the wavelength dependency of $k$ ($a_k$). (c) The dependency of $k$ on the backscatter fraction ($b$). The data points are colored by $a_k$. (d) The dependency of $k$ on the single-scattering albedo ($\omega$). The data points are colored by $a_k$.

## 5 Summary and conclusions

In this study, we presented a comparison of three different absorption photometers (AE31, PSAP, and MAAP), which measured ambient air at SMEAR II, a rural station located in the middle of a boreal forest in southern Finland. We also compared different correction algorithms that are used in determining the absorption coefficient ($\sigma_{abs}$) from the raw absorption photometer data. We studied how the algorithms affected the derived parameters and determined a multiple-scattering correction factor ($C_{ref}$) applicable at SMEAR II.

To obtain more reliable AE31 measurements, the AE31 data were compared against the MAAP data to acquire the $C_{ref}$ that is used in the processing of the AE31 data. Previous studies observed that $C_{ref}$ varied between different types of environments and stations, and here it was determined for the SMEAR II station, which represents the atmospheric conditions in a boreal forest. The resulting $C_{ref}$ values were 3.00, 3.13, 3.14, and 2.99 for the algorithms suggested by Weingartner et al. (2003), Arnott et al. (2005), Virkkula et al. (2007), and Collaud Coen et al. (2010), respectively. $C_{ref}$ determined at SMEAR II can be applied to other boreal forest sites as well, and even though the AE31 is an older model and no longer in production, the results can be used in post-processing older data sets or at sites that still operate the older AE31.

We also observed a clear seasonal cycle associated with $C_{ref}$, which was probably due to the variations in the optical properties of the aerosol particles, such as $b$ and $\omega$. We also observed some correlation between $C_{ref}$ and RH even though the RH in the instruments was kept below 40 %. These results show that the filter measurement methods seem to be rather sensitive to the RH even if the RH is below the recommended value of 40 %.

The results obtained for data corrected with the algorithm by Virkkula et al. (2007) were in many ways different from those obtained by Collaud Coen et al. (2010), who applied the Virkkula et al. (2007) correction to data from several stations in Europe. They found that the compensation parameter ($k$) used in the algorithm was highly nonstable and that it led to large outliers. They correctly stated that the difficulty of applying this correction is due to the naturally high variability in $\sigma_{ATN}$ as a function of time, which is for most of the time greater than the $\sigma_{ATN}$ decrease induced by filter changes. We therefore calculated 14 d running-average compensation parameters ($\pm 7$ d around each filter spot) in order to minimize these problems. The approach was obviously successful. It can be recommended that users of this method calculate running averages of $k$. The suitable period for the running average at each site depends on the rate of change in ATN, which determines how often the filter spots are changed. According to this study and to the study by Virkkula et al. (2015) the time period that includes about six to nine filter spot changes

https://doi.org/10.5194/amt-14-1-2021

on average seems to yield good results. At SMEAR II, a relatively clean site, this period was 14 d, and at SORPES, a rather polluted site, the period was 24 h.

The results showed a great variation between the $\alpha_{abs}$ derived from differently corrected $\sigma_{abs}$ data, and at SMEAR II the median $\alpha_{abs}$ for different algorithms varied in the range of 0.93–1.54. We also observed that most of the correction methods did not prevent the change in the wavelength dependency as the filter became more loaded, and therefore $\alpha_{abs}$ decreased notably with increasing attenuation (ATN). The correction algorithm by Virkkula et al. (2007) was the only AE31 correction algorithm that produced a stable $\alpha_{abs}$ for the increasing filter loading. For example, the $\alpha_{abs}$ derived from Aethalometer measurements is often used to describe the chemical properties of the particles and to describe the source of black carbon. Not taking the correction algorithm used and the effect of increasing filter loading into account could lead to the wrong interpretation of the results. According to our results, applying the Virkkula et al. (2007) correction algorithm could help resolve if the changes in $\alpha_{abs}$ were due to real variation or due to increased filter loading.

In general, at SMEAR II, the effect of the filter loading on average did not seem to cause a major difference in the measured $\sigma_{abs}$. However, a strong effect of increased filter loading was seen in the derived parameter $\alpha_{abs}$, which should encourage researchers to apply a filter-loading correction to filter-based absorption data. Even though on average $\sigma_{abs}$ did not seem to be greatly affected by the filter attenuation (ATN), the filter-loading effect can have a great effect when studying shorter periods and, for example, different seasons, which also justifies applying a correction to the data. According to our study the correction algorithms by Virkkula et al. (2007) and Arnott et al. (2005) performed the best in taking the seasonal variations of the aerosol particles into account. Also, the algorithm by Virkkula et al. (2007) produced the most stable $\alpha_{abs}$ that did not depend on ATN, which was not the case for the other algorithms.

When applying a correction algorithm to AE31 data, it is important to report which algorithm, $C_{ref}$ values, and other coefficients were used to acquire the final data product since the algorithms can have a notable effect on the results, especially on $\alpha_{abs}$. Our results showed that in general, it is good practice to perform the analysis of AE31 data by using a few different correction algorithms to see if the results vary notably for different algorithms.

*Code availability.* The MATLAB scripts, which were used to produce the figures and tables, are available from the authors. CE1

*Data availability.* The data collected from the SMEAR II is accessible through the Smart-MSEAR online tool (https://smear.avaa.csc. fi/, last access: 13 September 2021; Junninen et al., 2009).

*Supplement.* The supplement related to this article is available online at: https://doi.org/10.5194/amt-14-1-2021-supplement.

*Author contributions.* KL performed the data analysis and wrote the paper together with AV. AV and TP supervised the work of KL. PA and AV set up the aerosol optical measurements at SMEAR II. TP and MK designed the aerosol measurements at SMEAR II, wrote research proposals, and arranged funding for the research. All authors reviewed and commented on the paper.

*Competing interests.* The authors declare that they have no conflict of interest.

*Acknowledgements.* We thank the SMEAR II staff for taking care that all the measurements ran. We also thank the editor and the three anonymous reviewers whose comments improved this article.

Financial support of the University of Helsinki to ACTRIS-FI is gratefully acknowledged.

*Financial support.* This research has been supported by the European Union's Horizon 2020 research and innovation program via projects ACTRIS-2 (grant no. 654109) and iCUPE (grant no. 689443). Additional financial support was received through the Academy of Finland (Center of Excellence in Atmospheric Sciences) under the projects PROFI-3 (decision no. 311932), NanoBioMass (decision no. 307537) and decision no. 29664 TS2. This research was also supported by the Academy of Finland via project NABCEA (grant no. 296302) and by Business Finland via project BC Footprint (grant no. 528/31/2019).

Open-access funding was provided by the Helsinki University Library.

*Review statement.* This paper was edited by Saulius Nevas and reviewed by three anonymous referees.

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

**Remarks from the language copy-editor**

CE1    Please note slight edits to this section.

**Remarks from the typesetter**

TS1    Please note that the corrections of numbers are not language changes. If you still insist on changing these values in this section, the editor has to approve these changes. Please give an explanation of why this needs to be changed. Thanks.

TS2    To which project does this decision number belong?