# Peer review of "S1 The seasonal variation of $C_{\text{ref}}$"

_Atmospheric Measurement Techniques, 2020_

## Short Comment (SC1) · 3 Nov 2020

Dear authors,

I have found the paper very interesting. I would like to add a quick comment on section 4.3, where you discuss the Absorption Ångström exponent (AAE) retrieved when using different correction algorithms.

In the case of the AE31 (page 19, paragraph starting in line 29), I would like to mention that we have compared the AE31 with a multi-wavelength absorption photometer (see ref1), an instrument that works in a similar way to the MAAP but uses multiple

light sources at 5 different wavelengths. In our comparison we have found that the AAE retrieved from "raw" AE31 data was closer to the reference AAE, compared to the "corrected" AAE, calculated after correcting the data using two different correction algorithms.

References

(1) Massabò, D., Bernardoni, V., Bove, M. C., Brunengo, A., Cuccia, E., Piazzalunga, A., Prati, P., Valli, G., and Vecchi, R.:A multi-wavelength optical set-up for the characterization of carbonaceous particulate matter, J. Aerosol Sci., 60, 34–46, https://doi.org/10.1016/j.jaerosci.2013.02.006, 2013

(2) Saturno, J., Pöhlker, C., Massabò, D., Brito, J., Carbone, S., Cheng, Y., Chi, X., Ditas, F., Hrabě de Angelis, I., Morán-Zuloaga, D., Pöhlker, M. L., Rizzo, L. V., Walter, D., Wang, Q., Artaxo, P., Prati, P., and Andreae, M. O.: Comparison of different Aethalometer correction schemes and a reference multi-wavelength absorption technique for ambient aerosol data, Atmos. Meas. Tech., 10, 2837–2850, https://doi.org/10.5194/amt-10-2837-2017, 2017.

---

## Referee Comment (RC1) · Anonymous Referee #1 · 4 Nov 2020

This paper described results from the comparison experiments using three different light absorption filter-photometers, MAAP, PSAP, and Aethalometer, at a boreal forest site in Northern Europe. Correction of the output from these instruments has been considered one of the most important issues on the accurate determination of light absorption coefficient babs. In this study, authors conducted systematic comparison works to derive corrected babs from the measurements using three filter-photometers with different algorithms. The topics with which this paper deals meet the scope of Atmospheric Measurement Techniques (AMT); however, there are some points to be addressed before accepting the manuscript as an AMT paper. Please consider the following comments for the revision.

**Major comments**

1. Relative humidity of air for babs measurement by MAAP In this study, Cref was determined by the Equation (19). One of the bases of this way is the accuracy of  $\sigma$  abs, ref measured using the MAAP. In my reviewing process, I could not find very important related studies, for example Kanaya et al. (2013). In their study, BC concentrations measured using a MAAP (BCMAAP) were compared with those measured using a different filter photometer, COSMOS (Miyazaki et al., 2008). The dependency of MAAP sensitivity on relative humidity (RH) in MAAP has been discussed in relation to the changes in the optical properties of the glass filter tape (e.g., surface roughness). This change can be related to an increase in the surface roughness parameter to be used for the radiation transfer calculation (Petzold and Schönlinner, 2004) together with the RH. According to their studies, BCMAAP, namely  $\sigma$  abs, ref can be affected by RH in MAAP, even though the values of RH were lower than the recommended value (

P4-P5; The section 2.2 (instrument set-up) should be reorganized. The most important information is the set-up used in this study. So, the explanations about Fig 2 with the instrumental information should be describe as the basic experimental setup earlier than other information like the modification of the measurement flow line, the data availability, and the RH condition.

P11 L8-17; RH of air directed to the Nephelometer should be described in this section (2.4) to clarify the humidity condition of light scattering measurements and its impact on the hygroscopic growth of water-soluble aerosols.

P11 L19-21; Authors should describe why the difference in the size cut did not so greatly affect the results of the comparison experiments. Were there little impacts of (local) dust particles at the site?

P13 L29; The Cref values determined by different algorithms were described. Together with these values, their variabilities (e.g., 95% confidence interval) should be clarified here to show the statistical significance of the similarity and difference among correction algorithms. Statistical tests can help the discussion on the differences among variables.

P14 L14-16; It is hard for me to understand this explanation. This can only describe the possibility to describe one of the reasons of differences between CARN and CNC, and never account for the higher CARN than CNC. Please clarify the what this describes here. And again, without the significance of the differences, this kind of comparison works could not be established.

P16 L11; If the possible reasons of the lack of seasonal variations of CARN are added, authors can discuss the difference in the potential benefits of CARN compared to others (because the lack of seasonal variation is obviously beneficial). I believe that authors should discuss this point here to clearly differentiate the correction algorithms by their performance.
P17 L29-P18 L5; I am suspicious about how largely the particles can grow by water vapor at such low values of RH. Typical inorganic species never indicate large hygroscopic growth at RH

properly revise them.

References for the comments

Kanaya, Y., F. Taketani, Y. Komazaki, X. Liu, Y. Kondo, L. K. Sahu, H. Irie and H. Takashima (2013) Comparison of Black Carbon Mass Concentrations Observed by Multi-Angle Absorption Photometer (MAAP) and Continuous Soot-Monitoring System (COSMOS) on Fukue Island and in Tokyo, Japan, Aerosol Science and Technology, 47:1, 1-10, DOI: 10.1080/02786826.2012.716551.

Miyakawa, T., P. Mordovskoi and Y. Kanaya (2020) Evaluation of black carbon mass concentrations using a miniaturized aethalometer: Intercomparison with a continuous soot monitoring system (COSMOS) and a single-particle soot photometer (SP2), Aerosol Science and Technology, 54:7, 811-825, DOI: 10.1080/02786826.2020.1724870.

Miyazaki, Y., Kondo, Y., Sahu, L. K., Imaru, J., Fukushima, N. and Kanno, A. (2008) Performance of a Newly Designed Continuous Soot Monitoring System (COSMOS), J. Env. Monit., 10: 1195-1201. doi:10.1039/b80 6957c

Petzold, A. and M. Schonlinner (2004), Multi-Angle Absorption Photometry, A New Method for the Measurement of Aerosol Light Absorption and Atmospheric Black Carbon, J. Aerosol Sci., 35: 421-441.

---

## Referee Comment (RC2) · Anonymous Referee #2 · 9 Nov 2020

The authors present an interesting intercomparison of filter absorption photometers at a regional background site at Hyytiälä, Finland. The comparisons of this kind are important as a full characterization of the instrumental response is lacking and is compounded by the interwoven non-linearities of the measurement, which presents themselves as measurement artifacts, or as the authors' call this, systematic errors. The comparison of the MAAP, the AE31 and the PSAP partially addresses these shortcomings and presents new viewpoints on an urgent topic.

The manuscript fits well with the scope of AMT and can be accepted for publication after addressing the following major and specific comments.

[Figure]

The authors correctly point to the influence of the correction algorithm and its effective-ness on the slope of the inter-instrumental regression, which is used as the multiple scattering correction factor (Cref). The loading effect and the multiple scattering are artificially separated in the correction algorithms. Additionally, the particles, embedded in the filter, cause a known cross-sensitivity of the filter photometers to the scattering, which is explicitly described in the Arnott et al (2005) algorithm. Filter photometers also feature a dependence of the sensitivity on the location/depth of the particles in the filter matrix and are their sensitivity is therefore dependent no the size distribution of the sampled absorbing particles.

Weingartner et al. (2003), Park et al. (2010), Hyvärinen et al (2013), Segura et al (2014) and Drinovec et al. (2015) have discussed different approaches to showing the magnitude of these artifacts and their dependence on the loading of the sample spot. The authors should follow the same principle and plot the attenuation and absorption coefficients, and the absorption Angstrom exponent (AAE) as a function of the loading of the sample spot, for example as a function of ATN, Tr, ln(Tr). . . for all filter photome-ters in the study. This will also serve as a strong argument for using the MAAP as the reference.

Specific comments

Page 2, Lines 30 – P3, L5: There is another systematic error, not considered by the authors – the measurement of flow. The first issue is the reporting conditions of the flow: have they been unified across all instruments? If yes, please state the conditions in the respective Measurements and methods sections. The authors should include a word of caution for the instrumentation and the determination of the leakage – this is a multiplicative factor affecting the slope between instruments, which is (in the experience of the reviewer) often interpreted as being intrinsically instrumental.

P3, L 7-15: Please add the discussion on independent check of the correction algo-rithms with references to Park et al. (2010), Hyvärinen et al (2013), Segura et al (2014)

and Drinovec et al. (2015).

P4, L 16: Please add the widths of the different "wavelengths" in the filter photometers (for example from Müller et al., 2011).

P4, L 21: The reference to Fig. 1 is to a very nice picture of the experimental setup (which should remain in the manuscript) and not to the missing data availability plot. This missing figure could be added to the Supplement.

P4, L 21-28: It is RH change that perturbs the filter measurements, not RH per se. It would be interesting to take into account the RH change rate as well. For example, plot a companion to Fig. 4 with RH change rate, same for other instruments.

P4, L 30: Reference to Fig. 2 is in fact reference to Fig. 1.

P 5, L 8-10: Add the information on the filter material used.

P 5, L 17-18: This is incorrect. The intensities in PSAP and AE31 are normalized to the intensity measured under the clean part of the filter – the reference sample spot. This takes into account any possible drift in the LED intensities during the measurement period.

P5, L28: The sample spot should be measured. It changes with each spot slightly, especially due to leakage, when the filter tape is not well sealed. Was the correction for the differing values of A taken into account in this work?

P 6, L 6-12: The loss of sensitivity due to non-linear effects could be presented better. Please rewrite.

P 6, L 19-20: MAAP artifacts can be checked by a BC(ATN) plot, please see above. This justifies the use of the MAAP as the reference (further below, next paragraph).

P6, L 29: The authors talk about the precision here, not accuracy. Accuracy, however, is the parameter which is of importance. Please see above regarding the justification of the MAAP as the reference.

P 7, L 5: The unit-to-unit variabilities of different aethalometer types is very different - please expand and reference Müller et al. (2011) and Cuesta et al. (2020).

P 7 – 11, sections 2.3._ I disagree with the Anonymous Referee #1, these sections are important for understanding and interpretation the rest of the paper and should remain in the body of the manuscript.

P 8, L9: Please define single-scattering albedo as "omega".

P 8, L 13: Why linear dependency – compare Virkkula et al. (2007 and 2015).

P 8, L 13: Please define Angstrom exponent as "alpha_sca".

P 9, L 3, "... were calculated from...": Not clear if this relates to o the Hyytiälä measurements or to the Arnott et al. (2005). Please rephrase.

P 10, L 27: Which PSAP filter do these values relate to (Ogren et al., 2017)?

P 11, L 20-21: The averaging of PM1 and PM10 values is non-trivial due to possible regional contributions to BC in the larger size fractions. This does influence the non-linearities, which in-turn cause measurement artifacts that need to be corrected. The introduction mentions no change in the size of the sampled particles. The authors mention this briefly in section 4.1. Please add this information and provide an argument and discussion how this could influence the comparison.

P 13, L 10-11: Why calculate C_NC? It is loading dependent.

P 13, L 12-13: AAE is an absorption property, attenuation features loading effects, making AAE impossible to calculate, especially measurement at the lower wavelengths are heavily loading impacted. This paragraph needs to be extended and additional explanation on the determination of AAE provided.

P 13, L 20: Please extend the description of the fit – regression, it is not completely clear, cite Eq. 19...

P 13, L 21-22: The wavelength dependence of C is discussed in Bernardoni et al. (2020), which can be added to the discussion below (section 4.1, especially P 15, L24), provided it is calculated here.

P15, L 7-8: This is the place to discuss the influence of the correction algorithm performance on the C.

P 16, L 16-19: C_ref is the effective slope relative to the MAAP. Please add some discussion on the artifacts of all methods and their similarities/differences. What about size distribution artifacts? See also P17, L7.

P 16, L23-24: Or it describes the variation of the artifact better. Is this dependence on the parametrization scheme? Averaging?

P 17, L 7-8: This can be quantified, there are relevant measurements at Hyytiälä. Please provide this information.

P 17, L 14-17: This can also be described in a more quantitative manner, please see Virkkula et al. (2015) and Drinovec et al. (2017).

P 18, L 10: The intercept of the linear fit is the scattering artifact.

P 18, L 21: The data featuring low ATN is the one which features low loading artifacts and, therefore, a C with less uncertainty. This can be explored and the uncertainty as a function of the loading determined quantitatively.

P 18, L 27-29: The "smoothing" is site dependent and the non-corrected regression slope is always lower. The r2 of the non-corrected regression nis lower as well. Please discuss.

P 18, L 32: This is surprising, as one would expect that at low loading, the influence would be minimal. Is this a parametrization effect. Please discuss.

P 19, L 1-2: Please elaborate, the text is unclear.

[Figure]

P 19, L 18: This is different than explained above, Eq. 16. It is actually much more quantitative, as it allows the selectin of "good" AAE values by evaluating the fit r2, and ignoring the AAE values with low r2. This is used in French monitoring networks as a parameter to quality control the data and source apportionment of BC. Please use the r2 AAE selection and add this information in the manuscript.

P 19, L 31 – P 20, L 2: Please see above and Bernardoni et al (2020) and add to the discussion.

P 20, L 30: What was the maximum AE31 ATN for advancing the spot? Please add to the instrumental section.

P 20, L 31: Is this an observation of the data reported here (circular reference?) or an observation of Virkkula et al. (2007 and 2015) and Drinovec et al. (2017)?

P 20, L 32: This is not true. The correction algorithms take care of this. AAE dependence on ATN means that the loading correction is not working well. This is crucial as it shows that, except for V2007, the loading corrections do not function well! This is surprising, as this is the only correction not taking into account the cross-sensitivity to scattering. Why is "wavelength dependent k" better than other parameterizations? Same should be done for b_abs.

P 23, L 10-12: This depends on the rate dATN/dt, or the number of spots measured. This number can be counted and can be provided here. It is a good parameter for quality control and an important finding of the manuscript.

References

[revised manuscript text omitted]

––––––––––––––––––––––––

---

## Referee Comment (RC3) · Anonymous Referee #3 · 27 Nov 2020

The manuscript covers an important topic that has puzzled researchers for decades: the need to accurately measure light absorption by aerosol particles. Light absorption by aerosol particles are fundamental when assessing the direct radiative impacts of aerosols in the air but also on snow and ice. The work investigates how these measurements differ based on which post processing method is used in the quest to determine the absolute amount of light absorption by aerosol particles. The work covers three different filter based absorption photometers and how they compare against each other. The work further extends the analysis to cover how these post processing methods affects the spectral dependence of the light absorption coefficients and how this can lead to misleading conclusions when comparing one measurement to another

if not considering that the post processing method is of great significance.

General comments:

The Introduction would need a section where the goals of the study are clearly stated and then these goals should be addressed one by one in the conclusion section. This would help readers to grasp the extent of the research covered by the article.

The manuscript has dedicated a substantial proportion to the multiple scattering enhancement factor used in the Aethalometer post processing algorithms in the quest to make them perform better against the reference instrument MAAP. It is justified to scrutinize the multiple scattering enhancement factor of the Aethalometer but no attempt is made to scrutinize the multiple scattering enhancement in PSAP filters. PSAP filters are not as optically thick as the more rigid MAAP and Aethaloemter filters but multiple scattering is bound to occur in those filters too which would warrant a similar kind of investigation that is now presented for the AE31.

I wonder if the title of the manuscript couldn't be changed to something more inviting. The focus is on which effects different correction algorithms have on the post processed data which is an important topic indeed. Could the authors consider being more specific other than saying the manuscript deals with 'effects on different correction algorithms'. E.g. Effects of different correction algorithms on absorption photometers can lead to wrong interpretations if not . . . or something along those lines.

The English is generally good and it is easy to understand what the authors mean. There are however grammar errors that would need to be corrected and would improve the readability of the manuscript; e.g. definite articles and prepositions can be wrong or missing. In my specific comments I have made comments on those but the list is not exhaustive.

After addressing these comments and the specific comments below the manuscript is within the scope and of high enough scientific quality to be published in AMT. Please

do also consider the specific comments below for the revision.

Specific comments:

P1L16 resulted to –> resulted in

P1L20 filter measurements –> filter-based measurements

P2L8 climate in global –> climate on a global

P2L10: of the particles → of aerosol particles; scatter the light → scatter light

P2L11: "in color" is tautology so remove it

P2L12: suggest changing "light colored" to "bright"

P2L12: The sign of the radiative forcing is mentioned but could you be a bit more specific in what those signs actually mean i.e. write out cooling and warming instead of referring to the signs.

P2L16-19: sigma is a measure of light absorption and scattering, so it does more than "describe" it.

P2L24: I think that they are actually more unknown or not understood than actually defined.

P2L32: depends also → also depends

P2L34-P3L5: The discussion on Cref is focused on the different types of environments but does not address the fact that those studies cited weren't conducted in the same way. Some reference instruments were different than others which is likely to be a factor when comparing Cref values between studies E.g. a study using a MAAP as a reference instrument would yield different results compared to a study using a photoacoustic instrument as a reference measuring the same aerosols.

P3L8: correct "cast a so-called shadowing effect". Something casts a shadow but not a shadowing effect.

P3L10: remove "and determined coefficients"

P3L13-14: Here you could cite Collaud-Coen &al 2010 and Backman & al 2014 as those are relevant for what is claimed in the sentence.

P3L30: "remarkable" does not seem to be the correct word here

P4L6-9: Why mention CAPS if it is not used?

P4L10: Wouldn't the period be from Jun 2013 – March 2016 when all instruments are running? Why is then the period Jan 2012 – Dec 2017 chosen with the arguments of concurrent measurements?

P4L17: Remove the in 'measured the b_sca . . .'

P4L27: above accepted → above the accepted

P5L2-3: What is the Nephelometer actually measuring? The switch between PM1 and PM10 is done every 10 minutes and the flow through the comparatively large sensing chamber is 4.3 lpm. How fast is the Nephelomter flushed after a change in the inlet cut-size? It is not in seconds, but rather minutes as it does not flush evenly

P5L15 Bouguer is needlessly underscored.

P5L17-20 deltaT needs to be defined as the measurement interval.

P6L1 In the filter → In a filter

P6L7-8 It sounds like Weingartner is the cause of the "shadowing effect" when it is the filter and the particles that are the cause. Please rephrase.

P6L17-19: There isn't a correction algorithm for MAAPs but that does not mean that they don't need one. See e.g. Müller et al 2011 for e.g. the cross sensitivity to purely scattering aerosols as a function of filter loading.

P6L26: A radiative transfer scheme is no motivation for using the instrument as a reference instrument. The uncertainty and unit to unit variability (in that order) are arguments why it could be used as a "reference" although it does not provide the absolute truth either, as it is also filter based.

P6L28 'absorption instrument' sounds rather sloppy. Please use absorption photometer or something similar...

P7L1 remove 'again'

P7L2 'functional and popular' says who? Why not say widely used?

P7L6-7 Please be more specific than 'wavelength range is not as good'

P7L8 Problem for who? It can also be an advantage since it does not leak through the side of the filter tape.

P7L19-20 R can depend on other things too, not just ATN. R can be a function of single-scattering albedo, particle size, back scatter fraction etc. etc. This is the crux of the problem. Could be worth mentioning those things too.

P7L27-28 What were the criteria which lead you to choose these algorithms and not the others listed earlier? E.g. Schmid &al or Arnott&al are listed earlier but omitted here. Maybe they perform better and therefore warrants more investigation as more promising. You might want to state that if those were your criteria.

P8L11-16: Correct the grammar: e.g. remove articles before f, a and omega where not needed.

P8L16 resulted → resulting

P9L27-28: 14 days and filter changed on average once a day gives me 14 data points, not 9. The authors might want to rephrase a bit or write out the average filter change in days with a few decimals, like on average 1.55 days.

P10L8 Arnott &al 2005, not 2003.

P10L26 sigma_PSAP is not defined in the text.

P10L28 Shouldn't this equation be the Ogren 2010 ajusted equation as written out by Virkkula 2010 so that it reads sigma_ATN/(1.5557*Tr+1.0227). Or which equation did you use? The old Bond 1999 or the Ogren ajusted?

P11L5: Rephrase "we agreed the results"

P11L17: Which data did you use? PM1 or PM10? The uncertainty is greater for PM10 than for PM1 since the more signal is truncated when bigger particles are present.

P11L19 averaged for → averaged to

P11L25-26 concentration of the particles → concentration of particles, amount of the → amount of

P12L12 It is not a model but rather an equation that is used to make the source apportionment.

P12L13 Used for what? Just say that it is important measure of the aerosols ability to interact with light.

P12L25 less sensitive? The range for b is smaller than a_sca but how would it be less sensitive? I think you mean that the range is smaller. I suggest you remove this sentence as it is not relevant for the analysis in the manuscript.

P12L29 amount → amounts

P13L9 corrected by → corrected using

P13L18 within 1% limit → within a 1% limit

P13L12 why focus on WEI and COL when the biggest difference was to VIR?

P13L13 Similar effects? What effects I wonder? Do you mean average or mean concentrations? Being more precise would be more informative here.

P15L1-3 Would it be possible that the different Cref values in the mentioned studies is due to the reference instrument being something else than a MAAP?

P15L7 'describes' is not the correct word here

P15L15 real b_abs → true b_abs

P15L16 Which algorithms would be good if the reader is encouraged to use different algorithms based on their performance? At least you could state that e.g. the property derived from the AE31 should not depend on ATN after post processing. E.g. Fig. 8 shows clearly that some correction algorithms perform better than others when it comes to a_abs.

P15L20-31 Can the authors say something about which studies to trust and which ones not to trust?

P15L33 A linear fit does not average. Please rephrase.

P16L26 The sentence could need rephrasing. The fact that the Cref value changes is a strong indication that it is not a constant.

P17L3 'relatively more weight' could use rephrasing. How about saying that the optical size changes? This implies that smaller particles (Rayleigh reigime) aren't necessarily adding to the behaviour.

P17L13 correct: '. . . the relatively more the . . .'

P17L19-27 I understand that it can be hard to quantify the effect of RH on b_abs in this dataset but that is a very intresting topic. Based on your findings it appears that RH is more important than the aerosols single scattering albedo. Where was the RH measured? In the nephelometer? It is now shown in the schematics figure. How do you know that the observed RH dependence isn't from RH fluctuations in the MAAP sampling line which is not actively dried which then affects the CNC values as if that was something to do with the Aethalometer performance.

P19L21-27 The numbers mentioned in the text does not seem to match the figure. Please check if this is true or not. E.g. the lowest median a_abs value in the figure

does not seem to be 0.85 but rater close to 1 and the highest seems to be above 1.5 when in the text it is 1.48.

P19L32 measurements on → measurements at

P20L20 Figure 8 is only discussed here and is an excellent figure which I feel could be discussed a bit more. For example, the authors could for example use the figure as an illustration to state that whether it is b_abs or a_abs, the values should not depend on ATN and is an excellent test to check if the algorithm works. An important point to raise here would be that a_VIR seems to be the algorithms that performs the best.

P20L23 growed → grew or better still increased

P20L23 As a → In

P21L24 I suggest changing correlation to behaviour

P21L32-33 Please, rephrase the sentence

P22L7 remove 'about the'

P22L24 is the an model → is an old model

P22L28 You might want to mention that there are three different makers of tape for the AE33 and all of those have different Cref values.

P22L30 we observed also → we also observed

P23L24 Effect of increasing filter attenuation is sometimes called shadowing effect and sometimes filter loading effect. The authors should be consequent in what they call the effects.

Figures and tables

Table 1

Remove 'Also, ' and 'are presented in the table' To me it does not seem to be enough

to have the coefficeints in the table with only two digits. Three decimals would seem appropriate if possible.

Table 1 (which should read Table 2) These values are reported at the Aethalometer... → These values are reported at the MAAP...

Table 4 (should be Table 3) Remove 'the' from before k (a_k)

Figure 2 I am curious what the setup was like during the other years as this figure only illustrates the setup for 33% of the data.

Figure 3 The last sentence in the caption could say that the dashed line is the median for all the data as there are already other medians shown in the figure.

Figure 4 Adding Root Mean Square Errors could be a more quantitative way of expressing how well the instruments agree in addition to the correlation coefficient

Figure 6 Colored be → colored by

Figure 7 Please explain what the whiskers are in this figure as well. In the text it says that the statistics are as in Fig 5 but there is no boxplot there and in e.g. Fig 3 the mean is shown as an o whereas in this figure using an x.

Figure 8 and 9 Same thing here, explain the statistics of the boxplot.

The box plots could use some text about what the whiskers and boxes represent. For some figures it is a matter of a simple copy/paste.

Fig. 5 A better matrix for the performance of the vaious algorithms would be to include the Root Mean Square Error of the fits which would actually yield a quantitative value of the goodness of the fit in Mm-1. $R^2$ in all respect, but RMSE could be a good addition to the analysis.

---

## Author Comment (AC1) · 18 Jul 2021

**This paper described results from the comparison experiments using three different light absorption filter-photometers, MAAP, PSAP, and Aethalometer, at a boreal forest site in Northern Europe. Correction of the output from these instruments has been considered one of the most important issues on the accurate determination of light absorption coefficient $b_{abs}$. In this study, authors conducted systematic comparison works to derive corrected $b_{abs}$ from the measurements using three filter-photometers with different algorithms. The topics with which this paper deals meet the scope of Atmospheric Measurement Techniques (AMT); however, there are some points to be addressed before accepting the manuscript as an AMT paper. Please consider the following comments for the revision.**

**Major comments:**

**Relative humidity of air for $\sigma_{abs}$ measurement by MAAP:**

**In this study, $C_{ref}$ was determined by the Equation (19). One of the bases of this way is the accuracy of $\sigma_{abs,ref}$ measured using the MAAP. In my reviewing process, I could not find very important related studies, for example Kanaya et al. (2013). In their study, BC concentrations measured using a MAAP ($BC_{MAAP}$) were compared with those measured using a different filter photometer, COSMOS (Miyazaki et al., 2008). The dependency of MAAP sensitivity on relative humidity (RH) in MAAP has been discussed in relation to the changes in the optical properties of the glass filter tape (e.g., surface roughness). This change can be related to an increase in the surface roughness parameter to be used for the radiation transfer calculation (Petzold and Schönlinner, 2004) together with the RH. According to their studies, $BC_{MAAP}$, namely $\sigma_{abs,ref}$ can be affected by RH in MAAP, even though the values of RH were lower than the recommended value (<40%). I believe that authors should refer these papers in the discussion on the RH dependence of MAAP and discuss such uncertainty of MAAP related to RH condition. Some of conclusions, related to RH effect, should also be modified according to the discussion on the MAAP uncertainty.**

We added discussion with reference to these articles in the manuscript: *Kanaya et al. (2013) actually observed a slight dependency in the $\sigma_{abs}$ measured by MAAP so that at low RH (< 40 %) the $\sigma_{abs}$ increased with increasing RH, which is contrary to our results as we observed that MAAP observed relatively lower $\sigma_{abs}$ at higher RH. However, they also observed opposite behavior at higher RH (> 50 %). They suggested that the RH affects the surface roughness of the filter, which is used in the radiative transfer scheme (Petzold and Schönlinner, 2004), and therefore could affect the $C_{ref}$.*

**Readability:**

**Authors described the details of all the algorithms to correct the outputs of filter-photometers used in this study. I also believe that these descriptions are important, however, I, as one of readers, felt that the descriptions are somewhat lengthy because they are from previous studies, not originally from this study. To enhance the readability, I strongly recommend to reorganize the structure of the manuscript around the sections 2.3.1 and 2.3.2. The main part of these descriptions can be moved to Supporting Information or Appendix (which will be newly prepared in the revised manuscript). Only the essences (what types of**

**correction algorithms were used for AE31 and PSAP with proper references, what kinds of input parameters are needed for each algorithm, and so on) should be included in the main text.**

Here we received conflicting comments. We understand the recommendation of making the manuscript more compact. However, this time we decided to stick with the current structure because the algorithms are referred many times in the manuscript. We also thought that keeping the descriptions in the main manuscript prevents misunderstandings on what coefficients were used and how we used the algorithms in general.

**Specific comments:**

**P4-P5: The section 2.2 (instrument set-up) should be reorganized. The most important information is the set-up used in this study. So, the explanations about Fig 2 with the instrumental information should be describe as the basic experimental setup earlier than other information like the modification of the measurement flow line, the data availability, and the RH condition.**

The Sect. 2.2 was reorganized as recommended.

**P11 L8-17: RH of air directed to the Nephelometer should be described in this section (2.4) to clarify the humidity condition of light scattering measurements and its impact on the hygroscopic growth of water-soluble aerosols.**

We added the following text in Sect. 2.4: *Since scattering by aerosol particles is depends significantly on their size, the particulate light scattering is sensitive to hygroscopic growth. To prevent this, the integrating nephelometer operated with two Nafion-driers as shown in Fig. 1.*

**P11 L19-21: Authors should describe why the difference in the size cut did not so greatly affect the results of the comparison experiments. Were there little impacts of (local) dust particles at the site?**

As we discuss about absorption measurements with filter-based methods. When we correct the data depending on the ATN decreases because of both PM1 and PM10 and it is impossible to separate the effects of these in the loading correction. We added a sentence: *Since all the instrument measured the same sample air, combining the PM1 and PM10 data caused no discrepancies between the instruments.*

We also added discussion in the results: *Because it is impossible to separate the effect of different size cuts from a loaded filter, here the PM1 and PM10 measurements were combined and averaged together. In general, PM1 accounted for about 90% of the PM10 $\sigma_{abs}$; for the $\sigma_{sca}$ the fraction of PM1 was about 75%* (Luoma et al., 2019). Because absorbing particles, which is considered to consist mostly of black carbon, are typically in the fine mode (diameter < 1 µm), the $\sigma_{abs}$ is not expected to deviate much between the different size cuts. However, the differing size cuts, which causes more deviation in the $\sigma_{sca}$, could have affected the $\sigma_{abs}$ measurements since the particulate scattering causes apparent absorption and affect the multiple scattering in the filter. For example, the coarse particles (diameter > 1 µm) do not penetrate as deep in the filter as the fine mode particles, which could possibly influence on the Cref values. In an ideal situation the PM1 and PM10 absorption would have been measured by separate instruments.

**P13 L29: The $C_{ref}$ values determined by different algorithms were described. Together with these values, their variabilities (e.g., 95% confidence interval) should be clarified here to show the statistical significance**

**of the similarity and difference among correction algorithms. Statistical tests can help the discussion on the differences among variables.**

We added the confidence intervals in Table 2 as well as the following text in Sect. 4.1: *The results and their statistical variability are presented in Table 2. The relatively small standard error (SE) and the range of confidence interval (CI) indicate that the difference between the $C_{ref}$ values were statistically significant.*

**P14 L14-16: It is hard for me to understand this explanation. This can only describe the possibility to describe one of the reasons of differences between $C_{ARN}$ and $C_{NC}$, and never account for the higher $C_{ARN}$ than $C_{NC}$. Please clarify the what this describes here. And again, without the significance of the differences, this kind of comparison works could not be established.**

The Sect. 4.1 was reorganized so that first only the $C_{ref}$ values that were derived by linear regression ($C_{WEI}$, $C_{VIR}$, $C_{COL}$, and $C_{NC}$) were compared against each other. For these values the comparison was statistically justified. Since the $C_{ARN}$ was derived in a different manner, the $C_{ARN}$ is not necessarily comparable to the other $C_{ref}$ values.

**P16 L11: If the possible reasons of the lack of seasonal variations of $C_{ARN}$ are added, authors can discuss the difference in the potential benefits of $C_{ARN}$ compared to others (because the lack of seasonal variation is obviously beneficial). I believe that authors should discuss this point here to clearly differentiate the correction algorithms by their performance.**

To discuss more the advantages of both $C_{ARN}$ and $C_{VIR}$, we modified the paragraph in question in Sect. 4.1. The paragraph states now: *The seasonal variations for the $C_{ARN}$ and $C_{VIR}$ were less obvious than for $C_{WEI}$, $C_{COL}$, and $C_{NC}$. The lesser seasonal variation for the $C_{ARN}$ might be explained by the subtraction of the scattering fraction before the loading correction was applied and the $C_{ARN}$ was determined. For $C_{VIR}$, the lack of seasonal variation for was probably caused by the very strong seasonal variation of the compensation parameter (k; see Fig. 10a) as will be discussed below in Sect. 4.4. According to our results, the V2007 and A2005 accounted well the variations in the optical properties of the particles embedded in the filter and therefore the seasonal variations in the $C_{VIR}$ and $C_{ARN}$ were reduced.*

We also added the following statement in the conclusions: *According to our study the correction algorithms by Virkkula et al. (2007) and Arnott et al., (2005) performed the best in taking the seasonal variations of the aerosol particles into account.*

**P17 L29-P18 L5: I am suspicious about how largely the particles can grow by water vapor at such low values of RH. Typical inorganic species never indicate large hygroscopic growth at RH <40%, because their DRH are typically higher than 40% or so (even though considering the dehumidification process from higher RH condition). Furthermore, penetration depth of particles in filter is dependent on not only the particle size but also filter material properties and sampling flow rate (i.e., single fiber width, density of the fibers, and face velocity of air). The discussion here is highly speculative and fragmentary. Revisions to this discussion are strongly needed to better show precise interpretations.**

This part of the manuscript was reorganized and partly rewritten: *We observed slightly higher correlation (R = 0.30, p-value < 0.05) between the $C_{NC}$ and relative humidity (RH), which is presented in Fig. 4 (the correlation was similar for $C_{WEI}$ and $C_{COL}$, but weaker, about 0.09, for the $C_{VIR}$). Therefore, one possible reason for the observed seasonal variation of the different $C_{ref}$ values could be caused by changes in the instrumental RH and*

*the RH differences between the MAAP and AE31. The RH presented in Fig. 4 was measured in the MAAP and it varied between 5 – 40% since the periods when the filter of the MAAP was exposed for RH equal or larger than 40% were excluded from this study. Because the AE31 was equipped with Nafion-dryers, the RH in the AE31 varied less and was in the range of 5–20%. The RH can influence filter-based optical measurements by affecting to the optical properties of the aerosol particles and the filter fibers as well as by affecting the penetration depth of particles in the filter medium.*

*Due to the hygroscopic growth, the aerosol particles scatter more light in humid conditions compared to dry conditions. The enhanced scattering induced by higher RH could then increase the scattering and optical path in a particle-laden filter medium. However, at SMEAR II increasing RH should cause decreasing $C_{NC}$, since hygroscopic growth would increase the particulate scattering especially in the reference instrument MAAP. The hygroscopic growth may also affect the penetration depth of the particles in the filter (Moteki et al., 2010). When particles are penetrated deeper in the filter, the effect of the multiple scattering is higher increasing the measured $\sigma_{ATN}$. Because the RH in the MAAP was higher than in the AE31, the particles directed in the AE31 may have penetrated relatively deeper in the filter than the particles directed in the MAAP filter, in summer, larger difference in the RH between the instruments could increase the measured $C_{ref}$. However, the hygroscopic growth should not be significant in RH conditions below 40%, which is why the effects related to hygroscopic growth seem unlikely explanations.*

*Also, the optical properties of the filter may change if the filter is exposed to high RH conditions. The aerosol particles may take up water even below super saturation and when the liquid particles collide on the filter the moisture is taken up by the filter. Kanaya et al. (2013) compared the MAAP against Continuous Soot Monitoring System (COSMOS; Miyzaki et al., 2008) and actually observed a slight dependency in the $\sigma_{abs}$ measured by MAAP so that at low RH (< 40 %) the $\sigma_{abs}$ increased with increasing RH, which is contrary to our results as we observed that MAAP observed relatively lower $\sigma_{abs}$ at higher RH. However, they also observed opposite behavior at higher RH (> 50 %). They suggested that the RH affects the surface roughness of the filter, which then affects the scattering properties of the filter fibers.*

*The results show that even though we excluded the high RH data, the instruments seem to be sensitive to variations in RH even below the recommended 40%. However, the reason for the sensitivity remains unclear and would require more research and measurements and therefore further analysis is omitted from the scope of this article.*

**P18-P19 (sections 4.2 and 4.3): The performances of the correction algorithms as a function of *ATN* or *Tr* were evaluated in these sections. The slopes of $\sigma_{abs,AE31}$ (or $\sigma_{abs,PSAP}$)– $\sigma_{abs,ref}$ correlations and values of Absorption Ångström exponent $\alpha_{abs}$ were determined by the linear regression analysis. For better evaluations, it is beneficial to include the analyses of r2 values as a function of *ATN* and *Tr*. In terms of the measurement precision, *ATN* and *Tr* should be considered for quality control and quality assessment of the data obtained using filter-photometers. As an example, an evaluation of a miniaturized Aethalometer (AE51) in a previous study (Miyakawa et al., 2020) suggested that AE51 showed lower precision (i.e., lower r2) results in case of heavy aerosol loading on a collection filter (than not-used filter case).**

We added a table (Table 3) with the slopes and $R^2$ values for different *ATN* intervals and referred to this in the text. We also added discussion on the $R^2$ values: *According to the $R^2$ values presented in Table 3, the precision of*

*the AE31 decreased with increasing ATN. For example, the data corrected with the A2005 algorithm, the $R^2$ decreased from 0.96 for a clean filter (ATN < 20) to 0.90 for loaded filter (ATN > 60). However, the decrease in R2 was quite minor. Miyakawa et al. (2020) observed also rather high $R^2$ values between Aethalometer (model AE51) and a reference instrument (single particle soot photometer and COSMOS) when the ATN was below 70. When the ATN exceeded 70, the $R^2$ decreased more rapidly.*

**P19 L17: I believe that this sentence is not correct and not scientific (not -slope of a linear fit, simply slope of linear fit, because "-1" was multiplied in front of the slope term). So, this should be rephrased by using an equation or a proper expression.**

This was modified to: *The $\alpha_{abs}$ was determined as a linear fit over all the selected wavelengths according to Eq. (16).*

**P22 L25-28: These sentences should be included in discussion part, because they are not the actual outcome from this study.**

The text was moved to Sect. 4 Results and discussion.

**Captions of Figures 7, 8, 9: I think that "The explanation for the boxplots is the same as in Fig. 3" not Fig. 5. Furthermore, the marker types indicating the mean values are not always same for all figures (Figs. 3, 7, 8, 9). Please confirm the consistency and properly revise them.**

Fixed this.

---

## Author Comment (AC2) · 18 Jul 2021

**Author's response to RC2**
Manuscript: amt-2020-325

**Anonymous Referee #2**

**The authors present an interesting intercomparison of filter absorption photometers at a regional background site at Hyytiälä, Finland. The comparisons of this kind are important as a full characterization of the instrumental response is lacking and is compounded by the interwoven non-linearities of the measurement, which presents themselves as measurement artifacts, or as the authors' call this, systematic errors. The comparison of the MAAP, the AE31 and the PSAP partially addresses these shortcomings and presents new viewpoints on an urgent topic.**

**The manuscript fits well with the scope of AMT and can be accepted for publication after addressing the following major and specific comments.**

**The authors correctly point to the influence of the correction algorithm and its effectiveness on the slope of the inter-instrumental regression, which is used as the multiple scattering correction factor ($C_{ref}$). The loading effect and the multiple scattering are artificially separated in the correction algorithms. Additionally, the particles, embedded in the filter, cause a known cross-sensitivity of the filter photometers to the scattering, which is explicitly described in the Arnott et al (2005) algorithm. Filter photometers also feature a dependence of the sensitivity on the location/depth of the particles in the filter matrix and are their sensitivity is therefore dependent on the size distribution of the sampled absorbing particles.**

**Weingartner et al. (2003), Park et al. (2010), Hyvärinen et al (2013), Segura et al (2014) and Drinovec et al. (2015) have discussed different approaches to showing the magnitude of these artifacts and their dependence on the loading of the sample spot. The authors should follow the same principle and plot the attenuation and absorption coefficients, and the absorption Angstrom exponent (AAE) as a function of the loading of the sample spot, for example as a function of ATN, Tr, ln(Tr)… for all filter photometers in the study. This will also serve as a strong argument for using the MAAP as the reference.**

The figures of the $\sigma_{abs}$ against *ATN* and *Tr* are now presented in the supplementary material (Figs. S2 and S3) and we also included a table (Table 3 in the main text) that describes the performance of the AE31 and PSAP against MAAP for different filter loading intervals. At SMEAR II the MAAP was set to advance filter spot in 24 h interval and therefore, plotting the $\sigma_{abs}$ against the MAAP filter loading gives a false impression that the $\sigma_{abs}$ increases notably with increasing filter loading. This behavior is caused artificially with our data because the higher filter loading values are only reached when the $\sigma_{abs}$ is high (i.e., the higher the $\sigma_{abs}$, the more accumulation on filter within 24 h period). Because of the uninformativity and possible misconceptions, we did not plot the MAAP BC against the filter loading. The absorption Ångström exponent was already plotted against Tr and ATN in Figs. 7 and 8 (note new figure numbering).

We added/modified some text to discuss the new figures and table: *Surprisingly, the not corrected AE31 (Fig. 5e) did not seem to have a significant difference in correlation coefficient compared to for example to the data corrected with W2003 or CC2010 (Figs. 5a and d, respectively). However, the relation between the $\sigma_{abs,NC}$ and*

$\sigma_{ref}$ depend more on the ATN than for any filter loading corrected data, which is shown by the color coding (ATN) of the data points and in Table 3, which presents the slopes of the linear fits and $R^2$ values for different ATN intervals. If only data from highly loaded filter (ATN > 60 at 660 nm) were taken into account, the slopes of the linear fits were 0.97, 1.06, 0.99, 0.95, and 0.93 for W2003, A2005, V2007, CC2010, and not corrected (NC), respectively. The smallest decrease in the slope with increasing ATN determined for the loaded filter was observed for data that was corrected by V2007. Interestingly, the slopes for the loaded filter actually increased for data that was corrected by A2005, meaning that the $R_{ARN}$ had a relatively big effect with increasing ATN. The biggest decrease in the slope determined for a highly loaded filter was observed for the NC data, as expected. This observation underlines the need for filter loading correction if one studies shorter time periods. For longer time periods (e.g., trend analysis or studies of seasonal variation) the effect of the ATN smooths out, but for shorter time periods (e.g., case studies) the changing ATN can have a notable effect on the results if no filter loading correction is applied.

The dependency of the $\sigma_{abs}$ on the ATN and Tr is presented in supplementary material (Figs. S2 and S3). On average, the decrease in the $\sigma_{abs}$, which is not corrected for the filter loading ($\sigma_{abs,NC}$ and $\sigma_{abs,PSAP,ATN}$), with the increasing ATN and decreasing Tr is not clear. This effect is better seen in the results presented in Table 3. However, Fig. S3 shows that especially for the PSAP, the use of correction algorithms decreases the variation, which is a strong recommendation for using the correction algorihtms. This is also seen in the AE31 data, but the effect is less notable (Fig. S2).

According to the $R^2$ values presented in Table 3, the precision of the AE31 decreased with increasing ATN. For example, the data corrected with the A2005 algorithm, the $R^2$ decreased from 0.96 for a clean filter (ATN < 20) to 0.90 for loaded filter (ATN > 60). However, the decrease in $R^2$ was quite minor. Miyakawa et al. (2020) observed also rather high $R^2$ values between Aethalometer (model AE51) and a reference instrument (single particle soot photometer and COSMOS) when the ATN was below 70, but when the ATN exceeded 70, the $R^2$ decreased more rapidly.

**Page 2, Lines 30 – P3, L5: There is another systematic error, not considered by the authors – the measurement of flow. The first issue is the reporting conditions of the flow: have they been unified across all instruments? If yes, please state the conditions in the respective Measurements and methods sections. The authors should include a word of caution for the instrumentation and the determination of the leakage – this is a multiplicative factor affecting the slope between instruments, which is (in the experience of the reviewer) often interpreted as being intrinsically instrumental.**

We had forgotten to mention this in the original manuscript even though the flow calibrations were included in the data analysis. We have now added a paragraph describing the flow calibrations in the Sect. 2.2: *In order to perform a comparison study between the different absorption photometer, the sample flows of the instruments were calibrated. The sample flow of each instrument was regularly measured with a gillian flow meter and the flow reported by the instruments were corrected to match with the gillian measurements.*

**P3, L 7-15: Please add the discussion on independent check of the correction algorithms with references to Park et al. (2010), Hyvärinen et al (2013), Segura et al (2014) and Drinovec et al. (2015).**

Added: *In general, after correcting the data for the multiple scattering and loading effects, the absorption instruments agree rather well with the reference measurements (Drinovec et al., 2015; Hyvärinen et al., 2013; Park et al., 2010; Segura et al., 2014).*

**P4, L 16: Please add the widths of the different "wavelengths" in the filter photometers (for example from Müller et al., 2011).**

We added more description: *Here, we reported the typically used AE31 and PSAP wavelengths, which are reported in the AE31 manual and by Virkkula et al. (2005), respectively. These reported wavelengths deviate slightly from the ones measured and reported by Müller et al. (2011a) (see their Table 6). For the MAAP, we decided to use the wavelength reported by Müller et al. (2011a) since it rather commonly used and it clearly deviated from the wavelength reported by the manual.*

**P4, L 21: The reference to Fig. 1 is to a very nice picture of the experimental setup (which should remain in the manuscript) and not to the missing data availability plot. This missing figure could be added to the Supplement.**

We moved the availability plot in the supplement, so now the schematics of the set-up is Fig. 1.

**P4, L 21-28: It is RH change that perturbs the filter measurements, not RH per se. It would be interesting to take into account the RH change rate as well. For example, plot a companion to Fig. 4 with RH change rate, same for other instruments.**

We did study this. However, the change in $RH$ did not seem to affect the $C_{ref}$ at all. The correlation coefficient ($R$) between the $C_{NC}$ and $RH$ change (in units of %-points h$^{-1}$) was -0.03 and the $R$ between the $C_{NC}$ and relative change in $RH$ (units % h$^{-1}$) was -0.04. We added a statement about this in the manuscript: *The effect of rate of change in RH on the $C_{ref}$ was also studied, but there rate of change in RH did not show any correlation between $C_{NC}$.*

**P4, L 30: Reference to Fig. 2 is in fact reference to Fig. 1.**

Fixed this.

**P 5, L 8-10: Add the information on the filter material used.**

Added this information: *AE31 operated on quartz fiber filter (Pallflex, type 55619A), PSAP on quartz fiber filter (Pall, type E70-2075W), and MAAP on glass fiber filter (Thermo Scientific, type GF 10).*

**P 5, L 17-18: This is incorrect. The intensities in PSAP and AE31 are normalized to the intensity measured under the clean part of the filter – the reference sample spot. This takes into account any possible drift in the LED intensities during the measurement period.**

This was modified to: *$I_{t-\Delta t}$ and $I_t$ are the measured and normalized light intensities through the filter in the beginning of the measurement period $(t - \Delta t)$ and in the end of the measurement period $(t)$. The intensities are normalized by comparing them to the intensity measured through a reference spot. Normalizing the intensities accounts for the possible drifts and changes in the intensities of the LEDs.*

**P5, L28: The sample spot should be measured. It changes with each spot slightly, especially due to leakage, when the filter tape is not well sealed. Was the correction for the differing values of A taken into account in this work?**

Like for the flow correction, we forgot to describe the correction of filter spot size in the manuscript. This was applied already before, but now we added a description in the manuscript: *For the PSAP and AE31 we used the A values of 18.1 and 54.8 mm$^2$, which did deviate from the default ones that were 17.8 and 50.0 mm$^2$, respectively. The A used by default in the MAAP matched the measured one and therefore it was not corrected.*

**P 6, L 6-12: The loss of sensitivity due to non-linear effects could be presented better. Please rewrite.**

Rewrote this: *Absorbing particles induce a so-called "shadowing effect", which decreases the change in the intensity ($I_{t-\Delta t}I_t^{-1}$) as the filter gets more loaded (Weingartner et al., 2003). This means that the instrumental response is non-linear with increasing filter loading. The increasing filter loading has an opposite effect than the scattering of the filter fibers and particles: the absorbing particles collected on the filter decrease the optical path and therefore the reported $\sigma_{ATN}$ for a loaded filter is lower than for a pristine filter. This non-linearity is considered in the various correction algorithms presented in Sect. 2.3.1 and 2.3.2.*

**P 6, L 19-20: MAAP artifacts can be checked by a BC(*ATN*) plot, please see above. This justifies the use of the MAAP as the reference (further below, next paragraph).**

As already answered before, this figure was not included in the manuscript even though it would be useful and justify the use of MAAP as you said. However, due to the filter change in every 24 h, the figure shows an artificial increase in the BC concentration with *ATN* (i.e., high *ATN* values are only reached when the concentration of BC is high), which does not represent the reality.

**P6, L 29: The authors talk about the precision here, not accuracy. Accuracy, however, is the parameter which is of importance. Please see above regarding the justification of the MAAP as the reference.**

Please see the reply to the latest comment. We modified the sentence a bit: *According to the uncertainty and unit-to-unit variability, the MAAP is the most precise instrument for monitoring $\sigma_{abs}$ and black carbon (BC) concentration, which is typically derived from $\sigma_{abs}$ measurements. Also, the backscattering measurements from the filter reduce the artefacts caused by the scattering aerosol particles and the filter loading effect making it a more accurate instrument.*

**P 7, L 5: The unit-to-unit variabilities of different aethalometer types is very different - please expand and reference Müller et al. (2011) and Cuesta et al. (2020).**

This was mentioned in the previous paragraph: *Müller et al. (2011) reported that the unit-to-unit variability of the PSAP, AE31 and MAAP were about 8%, 20% and 3%. It must be noted that the unit-to-unit variability is a lot smaller, about 2 % for the new AE33 model (Cuesta-Mosquera et al., 2021).*

**P 7 – 11, sections 2.3.1 and 2.3.2: I disagree with the Anonymous Referee #1, these sections are important for understanding and interpretation the rest of the paper and should remain in the body of the manuscript.**

We decided to keep the algorithm descriptions in the main manuscript and not to move them in the supplementary material.

**P 8, L9: Please define single-scattering albedo as "omega".**

Fixed this.

**P 8, L 13: Why linear dependency – compare Virkkula et al. (2007 and 2015).**

I'm not sure if I understand the comment. Neither of these articles discussed the WEI2003 algorithm. Also, the effect of using different type of interpolation method should only cause a minor difference in the data.

**P 8, L 13: Please define Angstrom exponent as "alpha_sca".**

Fixed this.

**P 9, L 3: "… were calculated from…": Not clear if this relates to o the Hyytiälä measurements or to the Arnott et al. (2005). Please rephrase.**

This was rephrased: *To calculate the $\tau_{a,fx}(\lambda)$, we used the same a power law function $\tau_{a,fx}(\lambda) = \tau_{a,fx,521}\cdot(\lambda/521\ nm)^{-0.754} = 0.2338\cdot(\lambda/521\ nm)^{-0.754}$. $\tau_{a,fx}$ as Virkkula et al. (2011) and the resulted values are presented in Table 1.*

**P 10, L 27: Which PSAP filter do these values relate to (Ogren et al., 2017)?**

The Pallflex ones that are now also mentioned in the manuscript,

**P 11, L 20-21: The averaging of PM1 and PM10 values is non-trivial due to possible regional contributions to BC in the larger size fractions. This does influence the nonlinearities, which in-turn cause measurement artifacts that need to be corrected. The introduction mentions no change in the size of the sampled particles. The authors mention this briefly in section 4.1. Please add this information and provide an argument and discussion how this could influence the comparison.**

As we discuss about absorption measurements with filter-based methods. When we correct the data depending on the ATN decreases because of both PM1 and PM10 and it is impossible to separate the effects of these in the loading correction. We added a sentence: *Since all the instrument measured the same sample air, combining the PM1 and PM10 data caused no discrepancies between the instruments.*

We also added discussion in the results: *Because it is impossible to separate the effect of different size cuts from a loaded filter, here the PM1 and PM10 measurements were combined and averaged together. In general, PM1 accounted for about 90% of the PM10 $\sigma_{abs}$; for the $\sigma_{sca}$ the fraction of PM1 was about 75%* (Luoma et al., 2019). Because absorbing particles, which is considered to consist mostly of black carbon, are typically in the fine mode (diameter < 1 µm), the σabs is not expected to deviate much between the different size cuts. However, the differing size cuts, which causes more deviation in the σsca, could have affected the σabs measurements since the particulate scattering causes apparent absorption and affect the multiple scattering in the filter. For example, the coarse particles (diameter > 1 µm) do not penetrate as deep in the filter as the fine mode particles, which could possibly influence on the Cref values. In an ideal situation the PM1 and PM10 absorption would have been measured by separate instruments.

**P 13, L 10-11: Why calculate $C_{NC}$? It is loading dependent.**

Sometimes the $\sigma_{abs}$ is derived from AE31 data without any loading correction by using only some estimation for the $C_{ref}$. For example, WMO/GAW recommends (report 227) to apply a $C_{ref}$ value of 3.5 and not to apply any filter loading correction to the data. This value can be compared against that. This was argued for earlier in the text: *Current recommendation by the WMO and GAW is to assume the R(ATN) unity for the AE31 and to use a $C_{ref}$ value of 3.5, which was determined by a comparison study of different AE31 instruments (WMO/GAW, 2016). Therefore, we also studied "not-corrected" AE31 data for which we did not apply any R(ATN) correction or particulate scattering reduction, but only the multiple scattering correction.*

**P 13, L 12-13: AAE is an absorption property, attenuation features loading effects, making AAE impossible to calculate, especially measurement at the lower wavelengths are heavily loading impacted. This paragraph needs to be extended and additional explanation on the determination of AAE provided.**

When *ATN* (or $\sigma_{ATN}$) is plotted against $\log(\lambda)$ the plot follows linear fit. Therefore, there should not be a problem in using Ångström exponent to interpolate the *ATN* to different wavelengths. In general, Ångström exponent describes the wavelengths dependency of any optical property that is linearly dependent on the $\log(\lambda)$.

**P 13, L 20: Please extend the description of the fit – regression, it is not completely clear, cite Eq. 19...**

Rephrased: *$C_{ref}$ was determined as the slope of a linear regression for the whole data set (linear fit for a loading corrected $\sigma_{ATN}$ vs. $\sigma_{ref}$ -plot.*

**P 13, L 21-22: The wavelength dependence of *C* is discussed in Bernardoni et al. (2020), which can be added to the discussion below (section 4.1, especially P 15, L24), provided it is calculated here.**

I am not sure if I understood this comment and I do not quite understand what calculation is referred here as Bernardoni et al., (2021) did not observe statistically significant wavelength dependency for the $C_{ref}$. We added discussion (including the reference to Bernardoni et al., 2021) in Sect. 4.1: *There are both studies where constant $C_{ref}$ has been used and studies where wavelength dependent $C_{ref}$ has been used. Others observed no significant dependency for the $C_{ref}$ on the wavelength (Backman et al., 2017; Bernardoni et al., 2021; Collaud Coen et al., 2010; Weingartner et al., 2003; WMO/GAW, 2016).*

**P15, L 7-8: This is the place to discuss the influence of the correction algorithm performance on the *C*.**

We added a section (Sect. 2.3.3) that describes the differences of the algorithms: *The W2003 algorithm only depends on the ATN, otherwise it applies constant values and it does not consider the scattering subtraction. The A2005 is not a function of ATN but it takes the filter loading into account by summing the $\sigma_{abs}$ of the accumulated particles on the filter spot. It does not assume a constant for the scattering reduction but determines the fraction from the wavelength dependency of $\sigma_{sca}$. The CC2010 algorithm is similar to A2005 in a sense that it also defines its own scattering reduction factor and determines the filter loading correction by taking into account the properties of the particles accumulated in the filter. The V2007 only depends on the difference between the last and first measurements of two filter spots and it assumes no constant coefficients. The B1999 algorithm relies heavily on constants that describe the dependency on the Tr, whereas the V2010 algorithm is an iterative process that depends on the $\omega$. Both B1999 and V2010 consider the scattering reduction with a coefficient.*

Then the performance of the different algorithms were discussed more in Sect. 4.2 (not in the section about $C_{ref}$) because we though a deeper discussion on the algorithm differences fits better in there.

**P 16, L 16-19: $C_{ref}$ is the effective slope relative to the MAAP. Please add some discussion on the artifacts of all methods and their similarities/differences. What about size distribution artifacts? See also P17, L7.**

The comment about the size distribution is answered below. Modified the sentences and added some discussion:
*Since the $C_{WEI}$, and $C_{COL}$ had a similar seasonal variation, it is unlikely that the seasonal variation observed for $C_{NC}$ was caused by the lack of filter loading correction. It is rather surprising, that there was a seasonal variation for the $C_{COL}$ as well. The algorithm by CC2010 considers the wavelength dependency of scattering and the ω of the accumulated particles. It is rather surprising that taking these parameters, which have a seasonal variation at SMEAR II (Luoma et al., 2019; Virkkula et al., 2011), does not seem to reduce the seasonality of $C_{ref}$. For example, the seasonal variations between the $C_{WEI}$ and $C_{COL}$ are surprisingly similar, even though we applied constant f values in W2003*

*The seasonal variations for the $C_{ARN}$ and $C_{VIR}$ were less obvious than for $C_{WEI}$, $C_{COL}$, and $C_{NC}$. The lesser seasonal variation for the $C_{ARN}$ might be explained by the subtraction of the scattering fraction before the loading correction was applied and the $C_{ARN}$ was determined. The fact that the $C_{ARN}$ has less data points than the other the $C_{ref}$ values, might also explain part of the less amplified seasonality. For $C_{VIR}$, the lack of seasonal variation for was probably caused by the very strong seasonal variation of the compensation parameter (k; see Fig. 10a) as will be discussed below in Sect. 4.4. The algorithm by V2007 does not assume any coefficients but depends only on the difference between the last and first measurements of the filter spots. Therefore, it seems to adjust to seasonal changes whereas the other algorithms apply coefficients. According to our results, the V2007 and A2005 accounted well the variations in the optical properties of the particles embedded in the filter and therefore the seasonal variations in the $C_{VIR}$ and $C_{ARN}$ were reduced.*

**P 16, L23-24: Or it describes the variation of the artifact better. Is this dependence on the parametrization scheme? Averaging?**

This dependency is not parametrized since the data is just "justified" over the filter spot change, that creates the automatic seasonality

**P 17, L 7-8: This can be quantified, there are relevant measurements at Hyytiälä. Please provide this information.**

We did a quick comparison against the volume mean diameter (VMD) and geometric mean diameter (GMD), which were calculated for the PM1 and PM10 size modes by using the DMPS and APS data (in a similar manner as in Luoma et al., 2019). However, the correlation coefficients between these parameters and the $C_{ref}$ were very low (< 0.05). We did not add any discussion about this test in the manuscript to avoid adding too much description about size distribution measurements, which were not in the focus here. The APS and DMPS measure the size distribution of the whole aerosol population. In this situation, measurements of the soot particle size would have been useful. Unfortunately, this kind of an instrument is not operating at SMEAR II.

**P 17, L 14-17: This can also be described in a more quantitative manner, please see Virkkula et al. (2015) and Drinovec et al. (2017).**

We do this later in the manuscript (Sect. 4.4).

**P 18, L 10: The intercept of the linear fit is the scattering artifact.**

We added: *As presented in Table 3, the linear fits for the AE31 and PSAP data against the reference did not have an intercept of zero. This could be caused by the scattering artifact and the fact that the correction algorithms fail to take the scattering artefact completely into account. The intercept is the smallest for the B1999-corrected PSAP data and the largest to AE31 data. A fraction of $\sigma_{sca}$ is subtracted in the AE31 algorithms by A2005 and CC2010. However, the data corrected with these algorithms still have a higher or similar intercept as with the not-corrected data and the data corrected by the W2003 and V2007 algorithms. Considering the intercept, the V2007-corrected data performs the best in the AE31 vs. MAAP comparison, which is slightly surprising, since it does not take the scattering subtraction into account. For the V2010-corrected PSAP data, the intercept is negative suggesting that the V2010 algorithm overestimates the apparent absorption by scattering particles.*

**P 18, L 21: The data featuring low *ATN* is the one which features low loading artifacts and, therefore, a *C* with less uncertainty. This can be explored and the uncertainty as a function of the loading determined quantitatively.**

Actually we tested determining the $C_{\mathrm{ref}}$ for different *ATN* limits and the effect on the uncertainty was minor. This is also seen in Table 3, which reports the $R^2$ of the linear regression for different *ATN* intervals. The $R^2$ decreases only slightly with increasing *ATN*.

**P 18, L 27-29: The "smoothing" is site dependent and the non-corrected regression slope is always lower. The r2 of the non-corrected regression nis lower as well. Please discuss.**

Modified this and added some text: *For longer time periods (e.g., trend analysis or studies of seasonal variation), the effect of the ATN on the variation smooths out, but for shorter time periods (e.g., case studies) the changing ATN can have a notable effect on the results if no filter loading correction is applied. However, not correcting for the filter loading effect, the precision of the instrument and the $\sigma_{abs}$ or BC concentration on average are reduced, which is why applying a filter loading correction on filter-based photometers is always recommended.*

**P 18, L 32: This is surprising, as one would expect that at low loading, the influence would be minimal. Is this a parametrization effect. Please discuss.**

There was probably a misunderstanding here (mixing of low *Tr* to low loading). The sentence/paragraph was rephrased: *Figure 6b shows that V2010 overestimated the $\sigma_{abs}$ especially when the loading was high (Tr was low) and the linear regression was 1.24.*

**P 19, L 1-2: Please elaborate, the text is unclear.**

Rephrased and elaborated: *Also, the B1999 overestimated the $\sigma_{abs}$ slightly, but in general it performed better in comparison with MAAP and the slope was 1.07 (Fig. 6a). The linear fits in Figs. 6a and b include all the data, but Table 3 presents the slopes of the linear fits for data with different Tr limits. It is actually recommended to use PSAP data with Tr > 0.7 and if only this data is taken into account, especially the data corrected with the V2010 algorithm performs much better and has a slope of 1.01, but also the slope for the data derived with the B1999 algorithm yields a smaller slope of 1.04. Unlike for the AE31, the loading on the filter does not seem to affect to the precision of the PSAP at all as the $R^2$ values do not decrease with the increasing loading (Table 3).*

**P 19, L 18: This is different than explained above, Eq. 16. It is actually much more quantitative, as it allows the selectin of "good" AAE values by evaluating the fit r2, and ignoring the AAE values with low r2. This**

**is used in French monitoring networks as a parameter to quality control the data and source apportionment of BC. Please use the r2 AAE selection and add this information in the manuscript.**

The wrong sentence was modified to: *The $\alpha_{abs}$ was determined as a linear fit over all the selected wavelengths according to Eq. (16).*

The selection of only good AAE values according to the $R^2$ of the fit is very interesting. However, unfortunatelly, due to time limitations, we could not conduct the evaluation of the fits according to their $R^2$ now. However, we did a quicker check on the $R^2$ values in general for majority of the 1 h averaged data (~99 %) the $R^2 > -0.90$. Therefore removing AAE values with low $R^2$ should not make a notable difference in the data analysis.

**P 19, L 31 – P 20, L 2: Please see above and Bernardoni et al (2020) and add to the discussion.**

Discussion was added: *This could be done with several MAAPs operating at different wavelengths, by measuring the particles suspended in the air by photoacoustic method (Kim et al., 2019), by a polar photometer (Bernardoni et al., 2021), or by a Multi-Wavelength Absorption Analyzer (MWAA; Massabò et al, 2013).*

**P 20, L 30: What was the maximum AE31 ATN for advancing the spot? Please add to the instrumental section.**

We added a paragraph about this in Sect. 2.3 Absorption measurements: *At SMEAR II, the MAAP advanced the filter spot automatically once per day in 24 h intervals. The AE31 also advanced the spot automatically when the ATN reached 120 at 370 nm wavelength. The PSAP filters were aimed to be changed every second day, but due to weekends and holidays, the filters were sometimes changed only after several days. On average the PSAP filters were changed once per three days.*

We also added here: *on average the filter changed when the ATN at 660 nm ≈ 90.*

**P 20, L 31: Is this an observation of the data reported here (circular reference?) or an observation of Virkkula et al. (2007 and 2015) and Drinovec et al. (2017)?**

Rephrased this: *The k is often larger for the shorter wavelengths, which means that the non linearity caused by the increased filter loading is relatively stronger at the shorter wavelengths (Drinovec et al., 2017; Virkkula et al., 2007; Virkkula et al., 2015), which was also observed by this study (discussed in the next chapter).*

**P 20, L 32: This is not true. The correction algorithms take care of this. AAE dependence on ATN means that the loading correction is not working well. This is crucial as it shows that, except for V2007, the loading corrections do not function well! This is surprising, as this is the only correction not taking into account the cross-sensitivity to scattering. Why is "wavelength dependent k" better than other parameterizations? Same should be done for b_abs.**

Rephrased this: *According to these results, the other algorithms but V2007 do not seem to account enough the wavelength dependency of the R(ATN).*

We also added figures of the effect of loading on the absorption (see my first reply to Anonymous referee #2).

**P 23, L 10-12: This depends on the rate dATN/dt, or the number of spots measured. This number can be counted and can be provided here. It is a good parameter for quality control and an important finding of the manuscript.**

We added a recommendation here: *The suitable period for the running average at each site depends on the rate of change in the ATN, which determines how often the filter spots are changed. According to this study and to the study by Virkkula et al. (2015) time period that includes about 6 – 9 filter spot changes on average seems to yield good results. At SMEAR II, a relatively clean site, this period was 14 days and at SOPRES, a rather polluted site, the period was 24 hours.*

---

## Author Comment (AC3) · 18 Jul 2021

**Author's response to RC3**
Manuscript: amt-2020-325

**Anonymous Referee #3**

**The manuscript covers an important topic that has puzzled researchers for decades: the need to accurately measure light absorption by aerosol particles. Light absorption by aerosol particles are fundamental when assessing the direct radiative impacts of aerosols in the air but also on snow and ice. The work investigates how these measurements differ based on which post processing method is used in the quest to determine the absolute amount of light absorption by aerosol particles. The work covers three different filter based absorption photometers and how they compare against each other. The work further extends the analysis to cover how these post processing methods affects the spectral dependence of the light absorption coefficients and how this can lead to misleading conclusions when comparing one measurement to another if not considering that the post processing method is of great significance.**

**General comments:**
**The Introduction would need a section where the goals of the study are clearly stated and then these goals should be addressed one by one in the conclusion section. This would help readers to grasp the extent of the research covered by the article.**

We now combined the aims in to one paragraph in the intro section: *This study has two aims, which address the variation of the $C_{ref}$ and the differences between the different correction algorithms. The first aim is to provide a $C_{ref}$ value suitable for a boreal forest site and to study how the $C_{ref}$ varies between different correction algorithms. The second aim is to present how the different correction algorithms of the $\sigma_{abs}$ affect the measured optical properties of the particles.*

**The manuscript has dedicated a substantial proportion to the multiple scattering enhancement factor used in the Aethalometer post processing algorithms in the quest to make them perform better against the reference instrument MAAP. It is justified to scrutinize the multiple scattering enhancement factor of the Aethalometer but no attempt is made to scrutinize the multiple scattering enhancement in PSAP filters. PSAP filters are not as optically thick as the more rigid MAAP and Aethalometer filters but multiple scattering is bound to occur in those filters too which would warrant a similar kind of investigation that is now presented for the AE31.**

This indeed would be an interesting study to add in the manuscript and otherwise we would follow this recommendation and determine the Cref value also for the PSAP. Unfortunately, now the main author simply does not have time to conduct this study, since it would require quite a lot more of data analysis and rewriting the current scripts. To cover this comment, we did added discussion of the topic in the results: *If all the data was included in the comparison, as in Figs. 6a and b, the overestimation of $\sigma_{abs}$ would suggest to derive the $C_{ref}$ values also for the PSAP data. Here, we did not derive the $C_{ref}$ values for the PSAP, since they are not typically used in a similar way as for deriving the $\sigma_{abs}$ from the AE31 measurements. In general, the multiple scattering does not cause such a big artefact in filter material typically used in PSAP compared to the more thicker AE31 filters. However, if we considered only the data below Tr < 0.7 the PSAP and MAAP agree well for both correction algorithms. This result then suggests that there is no need for deriving a new $C_{ref}$ for PSAP. Svensson et al. (2019)*

*studied the multiple scattering in quartz filters and they derived the equations that can be used in determining the $C_{ref}$ value for PSAP. Differently to AE31 correction algorithms, the $C_{ref}$ used in PSAP algorithms is included in the coefficients of Eqs. 13 – 15 and therefore determining the $C_{ref}$ for PSAP is not as straightforward.*

**I wonder if the title of the manuscript couldn' be changed to something more inviting. The focus is on which effects different correction algorithms have on the post processed data which is an important topic indeed. Could the authors consider being more specific other than saying the manuscript deals with 'effects on different correction algorithms'. E.g. Effects of different correction algorithms on absorption photometers can lead to wrong interpretations if not… or something along those lines.**

We now at least rearranged the title so that the main thing, which is the effects of different algorithms, is first.

**The English is generally good and it is easy to understand what the authors mean. There are however grammar errors that would need to be corrected and would improve the readability of the manuscript; e.g. definite articles and prepositions can be wrong or missing. In my specific comments I have made comments on those but the list is not exhaustive.**

Thank you for putting so much effort in improving the language! We combined all the comments of language suggestions at the end of this document in order to keep the answers to bigger comments more easy to read.

**After addressing these comments and the specific comments below the manuscript is within the scope and of high enough scientific quality to be published in AMT. Please do also consider the specific comments below for the revision.**

**Specific comments:**

**P2L12: The sign of the radiative forcing is mentioned but could you be a bit more specific in what those signs actually mean i.e. write out cooling and warming instead of referring to the signs.**

We added: *i.e., negative sign for the cooling effect and positive sign for the warming effect* in parenthesis in the text.

**P2L16-19: sigma is a measure of light absorption and scattering, so it does more than "describe" it.**

"Describes" was changed to "is a measure".

**P2L24: I think that they are actually more unknown or not understood than actually defined.**

We rephrased this to give an angle that the correction algorithms for the nephelometer are systematically used and well accepted by the users, which is different to for example AE31 algorithms. *Correction algorithms and factors that minimize the error sources and uncertainties of nephelometer measurements are systemically used (Anderson and Ogren, 1998; Müller et al., 2011b).*

**P2L34-P3L5: The discussion on Cref is focused on the different types of environments but does not address the fact that those studies cited weren't conducted in the same way. Some reference instruments were different than others which is likely to be a factor when comparing Cref values between studies E.g. a study using a MAAP as a reference instrument would yield different results compared to a study using a photoacoustic instrument as a reference measuring the same aerosols.**

We added a note: *Since there is no generally accepted method for deriving the $C_{ref}$ values, the methods between different studies vary, which can also affect the results. In this study, we derived the $C_{ref}$ by comparing the AE31 measurements against another optical filter-based instrument.*

We also paid attention in describing the method used in the different $C_{ref}$ values that are referred in the article.

**P3L8: correct "cast a so-called shadowing effect". Something casts a shadow but not a shadowing effect.**
Was rephrased: *When the filter is loaded with absorbing particles, the particle loading decreases the response of the instrument.*

**P3L13-14: Here you could cite Collaud-Coen &al 2010 and Backman & al 2014 as those are relevant for what is claimed in the sentence.**
References were added.

**P3L30: "remarkable" does not seem to be the correct word here**
Changed to "significant".

**P4L6-9: Why mention CAPS if it is not used?**
We removed the mention of CAPS from here but left a mention later in the text just to give an explanation on why it was included in the measurement line (Fig. 2).

**P4L10: Wouldn't the period be from Jun 2013 – March 2016 when all instruments are running? Why is then the period Jan 2012 – Dec 2017 chosen with the arguments of concurrent measurements?**
We now did some modifications in the text (*This period was selected to have at least two absorption instruments running in parallel.*) and also in the data (affected Figs. 4 and 5). In order to prevent any differences caused by the different time periods in the instrument comparison, in Sect. 4.2, only data from Jun 2013 – Feb 2016 was used. Also, in comparing the absorption Ångström exponents, only parallel data were used.

**P5L2-3: What is the Nephelometer actually measuring? The switch between PM1 and PM10 is done every 10 minutes and the flow through the comparatively large sensing chamber is 4.3 lpm. How fast is the Nephelomter flushed after a change in the inletcut-size? It is not in seconds, but rather minutes as it does not flush evenly**
There was a flushing time in the nephelometer measurements, for the absorption data the three first minutes were omitted from the data analysis. This is now described in the text: *To hinder the effect of changing inlets, the first minutes of measurements after the inlet switch were omitted. For the absorption instruments the first three minutes were omitted and for the integrating nephelometer the first five minutes were omitted.*

**P5L17-20 deltaT needs to be defined as the measurement interval.**
Was defined.

**P6L7-8 It sounds like Weingartner is the cause of the "shadowing effect" when it is the filter and the particles that are the cause. Please rephrase.**

Rephrased: *Absorbing particles induce a so-called "shadowing effect", which decreases the change in the intensity ($I_{t-\Delta t}I_t^{-1}$) as the filter gets more loaded (Weingartner et al., 2003).*

**P6L17-19: There isn't a correction algorithm for MAAPs but that does not mean that they don't need one. See e.g. Müller et al 2011 for e.g. the cross sensitivity to purely scattering aerosols as a function of filter loading.**

Added a note: *However, even though the MAAP was used as the reference here, it must be remembered that like all the filter-based photometers, also MAAP suffers from the cross sensitivity to purely scattering aerosol and therefore it does not the best reference instrument (Müller et al., 2011a).*

**P6L26: A radiative transfer scheme is no motivation for using the instrument as a reference instrument. The uncertainty and unit to unit variability (in that order) are arguments why it could be used as a "reference" although it does not provide the absolute truth either, as it is also filter based.**

Changed to: *Since the uncertainty and unit-to-unit variability of the MAAP was a lot smaller than for the PSAP and Aethalometer we used the MAAP as the reference instrument for measuring $\sigma_{abs}$.*

**P7L6-7 Please be more specific than 'wavelength range is not as good'**

Changed to *not as wide.*

**P7L8 Problem for who? It can also be an advantage since it does not leak through the side of the filter tape.**

This was rephrased: *The PSAP filters have to be changed manually by the user so the instrument is not the best option to deploy at a remote site, but then again the leakage through the filter tape is lesser than for the MAAP and AE31.*

**P7L19-20 R can depend on other things too, not just ATN. R can be a function of single-scattering albedo, particle size, back scatter fraction etc. etc. This is the crux of the problem. Could be worth mentioning those things too.**

Added: *The R can also depend on other factors, such as the ω, and some of the algorithms take also other parameters than ATN into account.*

**P7L27-28 What were the criteria which lead you to choose these algorithms and not the others listed earlier? E.g. Schmid et al or Arnott et al are listed earlier but omitted here. Maybe they perform better and therefore warrants more investigation as more promising. You might want to state that if those were your criteria.**

The selection of the algorithms was rather practical, those were the ones we had already had experience on and we had the scripts ready for most of them. We aimed to use the same algorithms as in CC2010, however, the algorithm by Schmid et al. (2006) was left out by oversight. The newer algorithms were also left out because we did not know about their existence back in the days when we started doing the analysis. Now, due to time limitations and in order to get the manuscript resubmitted, we did not have time to add these missing algorithms in the study.

**P9L27-28: 14 days and filter changed on average once a day gives me 14 data points, not 9. The authors might want to rephrase a bit or write out the average filter change in days with a few decimals, like on average 1.55 days.**

This was fixed.

**P10L26 sigma_PSAP is not defined in the text.**

The equation was modified (the $\sigma_{PSAP}$ is not actually mentioned at all anymore).

**P10L28 Shouldn't this equation be the Ogren 2010 ajusted equation as written out by Virkkula 2010 so that it reads sigma_ATN/(1.5557*Tr+1.0227). Or which equation did you use? The old Bond 1999 or the Ogren ajusted?**

This was now modified to this formula.

**P11L5: Rephrase "we agreed the results"**

Rephrased: *Here, the iteration was stopped one the change was less than 1%.*

**P11L17: Which data did you use? PM1 or PM10? The uncertainty is greater for PM10 than for PM1 since the more signal is truncated when bigger particles are present.**

Indeed, Sherman et al., (2019) defined that the fractional uncertainty for the PM10 $\sigma_{sca}$ was 9.2 % and for the PM1 $\sigma_{sca}$ 8 %. We modified the sentence to: *The fractional uncertainty of the integrating nephelometer for PM10 has been reported to be ± 9 % (Sherman et al., 2015).*

**P12L12 It is not a model but rather an equation that is used to make the source apportionment.**

Rephrased to: *The $\alpha_{abs}$ is typically used in a set of empirical equations, that approximate the source of black carbon (BC) (Sandradewi et al., 2008; Zotter et al., 2017).*

**P12L13 Used for what? Just say that it is important measure of the aerosols ability to interact with light.**

Rephrased: *The single-scattering albedo (ω), which describes how big fraction of the total light extinction ($\sigma_{abs}$ + $\sigma_{sca}$) is due to scattering …*

**P12L25 less sensitive? The range for b is smaller than a_sca but how would it be less sensitive? I think you mean that the range is smaller. I suggest you remove this sentence as it is not relevant for the analysis in the manuscript.**

Removed.

**P13L12 why focus on WEI and COL when the biggest difference was to VIR?**

The paragraph was modified to make it less focused on WEI and COL: *The smallest determined $C_{ref}$ value was $C_{NC}$, which was expected. Since the $\sigma_{ATN}$ decreases for a loaded filter and the filter loading correction was not applied, the $C_{NC}$ has to be smaller than for the corrected data. Since the values of the $C_{WEI}$ and $C_{COL}$ were almost the same, the result suggested that on average, the loading corrections $R_{WEI}$ and $R_{COL}$ had on average a similar effect on the data. The highest value was determined for the $C_{VIR}$, which suggests that on average, the value of the $R_{VIR}$ was the lowest (i.e., the effect of filter loading correction in V2007 was stronger).*

**P13L13 Similar effects? What effects I wonder? Do you mean average or mean concentrations? Being more precise would be more informative here.**

Added *on average* to the sentence.

**P15L1-3 Would it be possible that the different Cref values in the mentioned studies is due to the reference instrument being something else than a MAAP?**

Added: *In these studies, however, the reference instruments were not filter-based photometers like in our study and that can have a remarkable effect on the results.*

**P15L7 'describes' is not the correct word here**

Modified to: *This is also closer to our observations, which is explained by the fact that at SMEAR II, the observed soot particles are likely aged and coated since there are no significant local emission sources.*

**P15L16 Which algorithms would be good if the reader is encouraged to use different algorithms based on their performance? At least you could state that e.g. the property derived from the AE31 should not depend on ATN after post processing. E.g. Fig. 8 shows clearly that some correction algorithms perform better than others when it comes to a_abs.**

We added the following recommendation in the conclusion section: *According to our study the correction algorithms by Virkkula et al. (2007) and Arnott et al., (2005) performed the best in taking the seasonal variations of the aerosol particles into account. Also, the algorithm by Virkkula et al. (2007) produced the most stable $\alpha_{abs}$ that did not depend on the ATN, which was not the case for the other algorithms.*

**P15L20-31 Can the authors say something about which studies to trust and which ones not to trust?**

Not really. We added some discussion in the manuscirpt: *Because the results between the different studies vary, it is difficult to conclude whether the $C_{ref}$ is wavelength dependent or not. To study the wavelength dependency of $C_{ref}$, it would be ideal to use a photoacoustic (like in Kim et al., 2019) or $\sigma_{ext} - \sigma_{sca}$ -methods as the reference measurements, since they are independent from the filter artefacts.*

**P15L33 A linear fit does not average. Please rephrase.**

Rephrased: *The different $C_{ref}$ values were not only determined as a linear fit that took into account the whole time series.*

**P16L26 The sentence could need rephrasing. The fact that the Cref value changes is a strong indication that it is not a constant.**

Rephrased: *As indicated by the seasonal variation, the $C_{ref}$ is not a constant value, but it depends on the optical properties of the particles embedded in the filter.*

**P17L3 'relatively more weight' could use rephrasing. How about saying that the optical size changes? This implies that smaller particles (Rayleigh reigime) aren't necessarily adding to the behaviour.**

Rephrased: *For example, the size dependent b and $\alpha_{sca}$ reach their maxima in summer and minima in winter, which indicates that in summer the fraction of smaller particles increases.*

**P17L19-27 I understand that it can be hard to quantify the effect of RH on b_abs in this dataset but that is a very intresting topic. Based on your findings it appears that RH is more important than the aerosols single scattering albedo. Where was the RH measured? In the nephelometer? It is now shown in the schematics figure. How do you know that the observed RH dependence isn't from RH fluctuations in the MAAP sampling line which is not actively dried which then affects the C_NC values as if that was something to do with the Aethalometer performance.**

The *RH* was determined for the MAAP by using the ambient *RH* and *T* measurements. We now have also studied how the rate of change of the *RH* affected the $C_{ref}$, which showed no correlation whatsoever. Like the referee already mentioned, in the scope of this manuscript it is difficult to define why the variation in the *RH* would effect the derived $C_{ref}$.

**P19L21-27 The numbers mentioned in the text does not seem to match the figure. Please check if this is true or not. E.g. the lowest median a_abs value in the figure does not seem to be 0.85 but rater close to 1 and the highest seems to be above 1.5 when in the text it is 1.48.**

Fixed this.

**P20L20 Figure 8 is only discussed here and is an excellent figure which I feel could be discussed a bit more. For example, the authors could for example use the figure as an illustration to state that whether it is b_abs or a_abs, the values should not depend on ATN and is an excellent test to check if the algorithm works. An important point to raise here would be that a_VIR seems to be the algorithms that performs the best.**

Added a mention in the conclusions: *The correction algorithm by Virkkula et al., (2007) was the only AE31 correction algorithm, which produced a stable α_abs for the increasing filter loading....* and ... *According to our results, applying the Virkkula et al. (2007) correction algorithm could help solving if the changes in α_abs were due to real variation or due to increased filter loading.*

**P21L32-33 Please, rephrase the sentence**

Rephrased: *The sizes of the particles affects their scattering properties and also on their penetration depth in the filter that again could affect the k.*

**P22L28 You might want to mention that there are three different makers of tape for the AE33 and all of those have different Cref values.**

This paragraph was moved to results and we added: *The filter material in AE33 is Teflon-coated glass filter tape (Pallflex type T60A20), but also the "old" filter tape (Q250F) has been used with AE33 and the recommended $C_{ref}$ values to use with these filters are 1.57 and 2.14, respectively (Drinovec et al., 2015).*

**P23L24 Effect of increasing filter attenuation is sometimes called shadowing effect and sometimes filter loading effect. The authors should be consequent in what they call the effects.**

We used now the "loading effect" term.

**Table 1 Remove 'Also, ' and 'are presented in the table' To me it does not seem to be enough to have the coefficeints in the table with only two digits. Three decimals would seem appropriate if possible.**

The text was modified and the third digits were added in the table for those parameters it was possible (*a* and *f*).

**Figure 2 I am curious what the setup was like during the other years as this figure only illustrates the setup for 33% of the data.**

The changes in the measurement line are explained in Sect. 2.2 Instrument set-up: *Also, during this period there were only few changes in the measurement line: in March 2017 the MAAP flow was decreased from 18 lpm to 9 lpm and Nafion dryers were installed in front of MAAP; and in November 2017 one of the two Nafion dryers were removed in front of the Nephelometer.*

Otherwise, the set-up changed with the instruments (the MAAP was installed and the PSAP removed).

**Figure 3 The last sentence in the caption could say that the dashed line is the median for all the data as there are already other medians shown in the figure.**

Fixed this.

**Figure 4 Adding Root Mean Square Errors could be a more quantitative way of expressing how well the instruments agree in addition to the correlation coefficient**

We now replaced the correlation coefficient with the coefficient of determination.

**Figure 7 Please explain what the whiskers are in this figure as well. In the text it says that the statistics are as in Fig 5 but there is no boxplot there and in e.g. Fig 3 the mean is shown as an o whereas in this figure using an x.**

Fixed this.

**Figure 8 and 9 Same thing here, explain the statistics of the boxplot. The box plots could use some text about what the whiskers and boxes represent. For some figures it is a matter of a simple copy/paste. Fig. 5 A better matrix for the performance of the vaious algorithms would be to include the Root Mean Square Error of the fits which would actually yield a quantitative value of the goodness of the fit in Mm-1. R2 in all respect, but RMSE could be a good addition to the analysis.**

Fixed this. And the correlation coefficient was replaced by coefficient of determination.

**Language suggestions:**

**P1L16 resulted to –> resulted in**

**P1L20 filter measurements –> filter-based measurements**

**P2L8 climate in global –> climate on a global**

**P2L10: of the particles –> of aerosol particles; scatter the light –> scatter light**

**P2L11: "in color" is tautology so remove it**

**P2L12: suggest changing "light colored" to "bright"**

**P2L32: depends also –> also depends**

**P3L10: remove "and determined coefficients"**

**P4L17: Remove the in 'measured the b_sca…'**

**P4L27: above accepted –> above the accepted**

**P5L15 Bouguer is needlessly underscored.**

**P6L1 In the filter –> In a filter**

**P6L28 'absorption instrument' sounds rather sloppy. Please use absorption photometer or something similar: : :**

**P7L1 remove 'again'**

**P7L2 'functional and popular' says who? Why not say widely used?**

**P8L11-16: Correct the grammar: e.g. remove articles before f, a and omega where not needed.**

**P8L16 resulted –> resulting**

**P10L8 Arnott &al 2005, not 2003.**

**P11L19 averaged for –> averaged to**

**P11L25-26 concentration of the particles ! –>concentration of particles, amount of the –> amount of**

**P12L29 amount –> amounts**

**P13L9 corrected by –> corrected using**

**P13L18 within 1% limit –> within a 1% limit**

**P15L15 real b_abs –> true b_abs**

**P17L13 correct: '…the relatively more the…'**

**P19L32 measurements on –> measurements at**

**P20L23 growed –> grew or better still increased**

**P20L23 As a –> In**

**P21L24 I suggest changing correlation to behaviour**

**P22L7 remove 'about the'**

**P22L24 is the an model –> is an old model**

**P22L30 we observed also –> we also observed**

**Figure 6 Colored be –> colored by**

**Table 1 (which should read Table 2) These values are reported at the Aethalometer...  –> These values are reported at the MAAP...**

**Table 4 (should be Table 3) Remove 'the' from before k (a_k)**

These were all modified in the text.